# Understanding and seasonal forecasting of hydrological drought in the anthropocene

Xing Yuan[1], Miao Zhang[1,2], Linying Wang[1,2], and Tian Zhou[3]

[1]CAS Key Laboratory of Regional Climate-Environment for Temperate East Asia (RCE-TEA), Institute of Atmospheric Physics, Chinese Academy of Sciences, Beijing, 100029, China
[2]University of Chinese Academy of Sciences, Beijing, 100049, China
[3]Pacific Northwest National Laboratory, Richland, WA 99352, USA

*Correspondence to*: Xing Yuan (yuanxing@tea.ac.cn)

**Abstract.** Hydrological drought is not only caused by natural hydro-climate variability, but can also be directly altered by human interventions including reservoir operation, irrigation and groundwater exploitation, etc. Understanding and forecasting of hydrological drought in the anthropocene are grand challenges due to complicated interactions among climate, hydrology and human. In this paper, five decades (1961-2010) of naturalized and observed streamflow datasets are used to investigate hydrological drought characteristics in a heavily managed river basin, the Yellow River basin in North China. Human interventions decrease the correlation between hydrological and meteorological droughts, and make the hydrological drought respond to longer time scale of meteorological drought. Due to large water consumptions over the middle and lower reaches, there are 118%-262% increases in the hydrological drought frequency, up to eight-fold increases in the drought severity, 21-99% increases in the drought duration, and the drought onset becomes earlier. The non-stationarity due to anthropogenic climate change and human water use basically decreases the correlation between meteorological and hydrological droughts, reduces the effect of human interventions on hydrological drought frequency while increasing the effect on drought duration and severity. A set of 29-year (1982-2010) hindcasts from an established seasonal hydrological forecasting system are used to assess the forecast skill of hydrological drought. In the naturalized condition, the climate-model-based approach outperforms the climatology method in predicting the 2001 severe hydrological drought event. Based on the 29-year hindcasts, the former method has a Brier Skill Score of 11%-26% against the latter for the probabilistic hydrological drought forecasting. In the anthropocene, the skill for both approaches increases due to dominant influence of human interventions that have been implicitly incorporated by the hydrological post-processing, while the difference between two predictions decreases. This suggests that human interventions can outweigh the climate variability for the hydrological drought forecasting in the anthropocene, and the predictability for human interventions needs more attention.

## 1. Introduction

Drought is a natural phenomenon occurring due to climate variability that is associated with oceanic and/or terrestrial anomalies (Hong and Kalnay, 2000; Hoerling and Kumar, 2003). As the rainfall deficit reaches a certain threshold, the meteorological drought occurs. If the meteorological drought persists for a period of time, it will decrease the soil moisture

and river flow, resulting in agricultural and hydrological droughts. Although the rainfall deficit is a primary driver for the agricultural and hydrological droughts, terrestrial hydrological processes (e.g., snow melting, evapotranspiration), geological and topographic conditions also play a non-trivial role in the drought propagation (Van Loon et al., 2012; Rimkus et al., 2013; Teuling et al., 2013; Stoelzle et al., 2014; Staudinger et al., 2015). Therefore, monitoring and forecasting of agricultural and hydrological droughts not only provide more relevant guideline for the management of agricultural and water resources, but also raise challenging science questions on the mechanism and predictability of multi-scale drought processes (Pozzi et al, 2013; Yuan et al., 2013; Wood et al., 2015).

Given that rainfall deficit is a major cause for the hydrological drought, many studies focus on the understanding of propagation from meteorological to hydrological drought. For instance, Vicente-Serrano and López-Moreno (2005) investigated the relationship between streamflow and antecedent rainfall by using a correlation analysis, and found that hydrological drought index series have the highest correlation with a two-month accumulated meteorological drought index series in a mountainous Mediterranean basin. Such correlation analysis method was then widely used to understand the time scale of hydrological drought (Rimkuset al., 2013; Haslingeret al., 2014; Bloomfield et al., 2015; Folland et al., 2015; Niu et al., 2015; Kumar et al., 2016). A recent study from Barker et al. (2016) comprehensively investigated the relationship between meteorological and hydrological droughts for 121 near-natural catchments in the United Kingdom, where the relationship was found to be associated with natural climate and catchment properties.

Besides natural climate and hydrological processes that affect the development of hydrological drought, human activities such as land use and land cover change, irrigation, reservoir operation and groundwater exploitation can also influence hydrological drought significantly (AghaKouchak et al., 2015; Van Loon et al., 2016a). López-Moreno et al. (2009) found that the second largest reservoir in Europe increased the duration and severity of hydrological drought over downstream areas. Similar studies found that reservoir regulation might reduce the drought severity over upstream areas but increase it over downstream areas over Australian and Chinese catchments (Wen et al., 2011; Zhang et al., 2015), as many reservoirs were built for meeting the irrigation demand more reliably. However, there is limited knowledge on the integrated impact of human activities on hydrological drought processes over a large river basin due to the lack of human water use data, which is one of the major issues that hinders the understanding of hydrological drought in the anthropocene (Van Loon et al., 2016b). An alternative approach is to use a land surface hydrological model (Wada et al., 2013; Zhou et al., 2016) or a less complicated water balance model to recover the naturalized streamflow by assimilating the reported water use data. The naturalized streamflow data can be used to calibrate hydrological model without human components to investigate natural response of hydrological processes to the climate variations (Yuan et al., 2016), it can also be used to investigate the integral anthropogenic impact on hydrological drought by comparing with observed streamflow time series.

The ultimate goal of the understanding of drought processes is to facilitate the development of drought early warning systems for drought adaptation and mitigation. Fortunately, the development of ocean-atmosphere-land coupled general circulation models (CGCMs) provides an unprecedented opportunity to transfer advances in seasonal forecasting research (Kirtman et al., 2014; Yuan et al., 2015a) into an integrated drought service. Besides meteorological drought forecasts (Dutra

et al., 2012; Yuan and Wood, 2013; Ma et al., 2015), agricultural drought forecasts with dynamical seasonal climate forecast models have also been widely applied and evaluated (Luo and Wood, 2007; Mo et al., 2012; Sheffield et al., 2014; Yuan et al., 2015b; Thober et al., 2015). However, dynamical forecasting of hydrological drought based on the CGCM-hydrology coupled approach (Yuan et al., 2015a) has received less attention (Trambaueret al., 2015; Sikder et al., 2016), although there are many statistical forecasting studies for low flows. The reasons are threefold: 1) a skilful seasonal forecasting of streamflow usually occurs over basins with large storages of snow, surface and/or subsurface water (Wood and Lettenmaier, 2008; Koster et al., 2010), and strong control from initial hydrological conditions limits the added value from climate predictions (Wood et al., 2016; Yuan, 2016); 2) unlike meteorological drought forecasts, both agricultural and hydrological drought forecasts are influenced by the uncertainty in hydrological model, and the hydrological drought forecasting tends to be more challenging since the errors from upstream areas can be transferred to or even amplified in downstream areas; and 3) many river basins are altered by human activities, where the management impacts are often neglected in most dynamical forecasting system. In fact, seasonal forecasting of hydrological drought in the anthropocene raises the questions of how to define the predictability of the anthropogenic processes within a coupled hydro-climate system, how to distinguish the uncertainty from each component, and how to assess the forecast skill of hydrological drought with natural and anthropogenic forcings.

This paper focuses on the understanding and seasonal forecasting of hydrological drought over a heavily managed river basin in North China, the Yellow River basin. Both naturalized and observed streamflow along the mainstream of Yellow River will be used to investigate the relationship between meteorological and hydrological droughts under natural and anthropogenic conditions, to quantify the influence of human activities on the characteristics of hydrological drought (e.g., drought frequency, duration and severity, and seasonality of hydrological drought onset), and to assess hydrological drought forecasting in the anthropocene with an experimental seasonal hydrological forecasting system established over Yellow River basin (Yuan et al., 2016).

2.  **Data and Method**

**2.1 Study domain and hydroclimate observation data**

Precipitation dataset at 0.25-degree resolution during 1961-2010 was interpolated from 324 meteorological stations within Yellow River basin (Yuan et al., 2016). Figure 1 shows that regional mean annual precipitation decreases from southeast to the northwest. Most precipitation over Yellow River occurs in summer season due to the influence of East Asian monsoon, resulting in a strong seasonality of precipitation, with more than 80% of annual precipitation falls within May-September (Fig. 2). For each hydrological gauge, the sub-basin mean precipitation was calculated to investigate the relationship between meteorological and hydrological droughts.

Figure 1 shows locations of 12 mainstream hydrological gauges used in this study, with Tangnaihai gauge in the headwater region and Lijin gauge at the outlet of entire Yellow River basin that has a drainage area of $7.52 \times 10^5$ km$^2$. Details of the drainage areas for the 12 gauges can be found in Yuan et al. (2016). Both natural and observed streamflow datasets at

monthly time scale during 1961-2010 were provided by local authority of Yellow River. The naturalized streamflow ($W_{nat}$) was calculated as follows:

$$W_{nat}=W_{obs}+W_{irr}+W_{idu}+W_{civ}+W_{div}+W_{res},$$

where $W_{obs}$ is the observed streamflow; $W_{irr}$ is surface water consumed in irrigation, i.e., irrigated surface water that is transpired from crop, evaporated from bare ground and river/channel water surface, absorbed into soil (through infiltration, percolation and recharge to shallow groundwater), leaked from river/channel bed to groundwater during transportation, so the $W_{irr}$ is not simply those withdrew from river, while it is actually the surface water consumption in the processes of irrigation that already considers the flow returned to the river; similarly, $W_{idu}$ and $W_{civ}$ are surface water consumed by industrial and civil sectors, by considering the waste water returned to the river; $W_{div}$ is the interbasin water diversion, and $W_{res}$ is the water regulated by reservoirs. To account for the non-stationarity both from anthropogenic climate change (Milly et al., 2008, 2015) and land use/land cover changes (Villarini et al., 2009; Zhang et al., 2011), the variable runoff coefficient (annual runoff divided by annual precipitation) and more physically based methods that consider the soil conservation (e.g., Grain for Green project over Loess Plateau), effect of groundwater overdraft on surface water, and evaporation loss from reservoirs or channels, are used to correct the naturalized streamflow. Such correction is critical over Yellow River due to significant climate change and human interventions (Zhang et al., 2011). For example, small, medium and heavy rainy days decreased by 20-26% over the areas between Hekouzhen and Longmen gauges in 1980s as compared with those in 1950s and 1960s, with duration of extremely heavy rainfall decreased from 0.82 day to 0.38 day. The annual streamflow over the middle reach of Yellow River could decrease by 19% with the same annual precipitation due to ecological conservation, and could decrease by 47% due to the riverbed leakage induced by groundwater overdraft. The corrections for the annual mean streamflow over the areas from Tangnaihai to Lanzhou, and down to Hekouzhen, Longmen, Sanmenxia, Huayuankou and Lijin are $3.16\times10^8$, $0.80\times10^8$, $6.17\times10^8$, $10.99\times10^8$, $9.23\times10^8$ and $2.61\times10^8$ m$^3$, respectively. Definitely, there are uncertainties in the naturalized streamflow which is difficult to quantify because of absent "real" streamflow under natural conditions over Yellow River. But to our knowledge, this is the most comprehensive estimation of naturalized streamflow over Yellow River basin so far, based on abundant data from different sectors. Yuan et al. (2016) used the naturalized streamflow to calibrate the Variable Infiltration Capacity (VIC) land surface hydrological model (Liang et al. 1996) during 1961-1981, when the human interventions were supposed to be limited. The naturalized streamflow was then compared with VIC simulations during the validation period 1982-2010, and the naturalized streamflow agreed with VIC simulations quite well with Nash–Sutcliffe efficiency (NSE) varied between 0.71-0.91 (Figure 4 in Yuan et al., 2016).

Except for the Tangnaihai gauge in the headwater region, streamflow at the hydrological gauges used in this study was influenced by human interventions (Fig. 3). Therefore, the Yellow River is an ideally large river basin to investigate the hydrological drought processes and predictability in the anthropocene. In general, human interventions decrease streamflow over upper and middle reaches of Yellow River during rainy season while increase it during dry season (Figs. 3b-3h). This suggests that reservoirs in the upper and middle reaches of Yellow River store rain water in wet season and distribute it in the remaining time of the year according to the need, which is similar to regulations in other parts of the world (Wada et al.,

2014). Actually, Figure 4a shows that the annual mean observed streamflow at upper reaches can be higher than the naturalized streamflow during dry years due to the reservoir water release (e.g., years 2000, 2002, 2006 and 2010 for Lanzhou gauge). Over the lower reaches, the observed streamflow is significantly lower than the naturalized streamflow during wet season due to heavy water consumption, and riverbed leakages because of groundwater overdraft and possible geomorphology change caused by sediment accumulation (Figs. 3i-3l). The observed streamflow is close to the naturalized streamflow during dry season because of no significant water consumption or reservoir management. Figure 4 also shows that the magnitudes of reservoir storage changes are quite small as compared with streamflow. In fact, the mean absolute changes of reservoir storage during 1998-2010 are about 14%-38% and 12%-14% of observed and naturalized streamflow, respectively. This suggests that other human interventions, such as direct withdrawal of surface water for agricultural, industrial and civil consumptions, account for a large part of streamflow variations over Yellow River.

## 2.2 Definitions of drought indices and hydrological drought event

The Standardized Precipitation Index (SPI; McKee et al., 1993) was used as the meteorological drought index. The main advantage of SPI is its multiscale nature, i.e., it can be used to represent meteorological drought at different time scales. In this study, sub-basin mean precipitation datasets averaged over antecedent 1 to 24 months were used to calculate SPI-1 to SPI-24. To account for the seasonality, the SPI for each target month and for each time scale was calculated separately by using 50-year data. For example, SPI-6 at October 1982 was calculated by firstly fitting an empirical distribution based on the precipitation averaged between May and October during 1961-2010, and the May-October mean precipitation in 1982 was then used to determine the SPI6 value for October 1982.

Similarly, monthly naturalized streamflow datasets at 12 hydrological gauges during 1961-2010 were also standardized by using the same procedure, resulting in hydrological drought index named as Standardized Streamflow Index (SSI). Note that SSI was similar to the standardized runoff index (SRI) defined by Shukla and Wood (2008), except that streamflow was used here for a standardization. For the anthropogenic streamflow, the same parameters of probabilistic distributions fitted from the naturalized streamflow were used, and the SSI values were then calculated. Both the 50-year (1961-2010) data and a series of 30-year transient climatology (e.g., 1961-1990 climatology was used for year 1975, 1962-1991 climatology was used for 1976, and so on) were used to calculate the stationary and non-stationary SPI/SSI values respectively. A threshold of -0.8 was used to represent a drought condition for both SPI and SSI. And a hydrological drought event was selected when the SSI was below -0.8 for at least 3 continuous months (Yuan and Wood, 2013), where the drought onset month was the first month where SSI fell below -0.8. Once a hydrological drought event occurred, both the duration months and severity ($\sum_{i=1}^{n}(-0.8 - SSI_i)$), where n is the number of month for the drought event) were calculated. And the number of drought events, mean drought duration and severity were obtained for both naturalized and observed SSI.

## 2.3 Seasonal hydrological ensemble hindcast datasets

A number of seasonal hydrological ensemble hindcast datasets created by Yuan (2016) were used in this study. To have this paper self-contained, the hindcast experiments were briefly described below. Firstly, a continuous offline hydrological simulation with calibrated VIC model and river routing model driven by observed meteorological forcings from 1951 to

2010, was conducted to generate the initial hydrological conditions (ICs) for the hydrological hindcasts (Yuan et al., 2016). The observed meteorological forcing datasets including daily precipitation, daily maximum and minimum surface air temperature, and surface wind were interpolated from 324 China Meteorological Administration stations. The VIC model version 4.0.5 was used to predict runoff in a water balance mode over the entire Yellow River basin with 1321 grid cells at

0.25-degree resolution, and a routing model was used to translate the runoff into streamflow at each 0.25-degree grid cell, and to route the flow into rivers and finally into the ocean (Yuan et al., 2016). Secondly, a set of 6-month Ensemble Streamflow Prediction (ESP) experiments were carried out by using the VIC and routing models, where the hydrological models were initialized with generated ICs and were forced by 28 ensemble forcing during 1982-2010 excluding target year. It is named as ESP/VIC hereafter. Thirdly, a grand ensemble of 99 realizations from eight North American Multimodel

Ensemble (NMME; Kirtman et al., 2014) models was used to force hydrological models to generate the NMME/VIC hindcast dataset (Yuan, 2016). Here, the 1-degree NMME global hindcasts of monthly precipitation and temperature were bilinearly interpolated into 0.25-degree, the interpolated monthly hindcasts for each NMME model were then bias-corrected independently against observations by using the quantile-mapping method (Wood et al., 2002) in a cross-validation mode (i.e., dropping observation and forecast in the target year when generating the climatology), and these bias-corrected monthly

hindcasts were finally temporally disaggregated to daily by historical sampling and rescaling (Yuan, 2016). The forecast streamflow can be directly compared with offline simulated streamflow. To compare with observed streamflow, a hydrological post-processing procedure (Yuan, 2016) was applied to adjust the forecast streamflow statistically by using the Bayesian theory. To account for the non-stationarity, the hydrological post-processing was carried out by using observed streamflow during 1982-2010 in a cross-validation mode (Yuan, 2016), and the corresponding SSI was also calculated by

using the concurrent hindcast period (i.e., 1982-2010). It should be pointed out that the non-stationarity could be reduced further in a real-time forecasting mode because of gradual use of concurrent climate and hydrology information for the calibration and initialization of hydrological model for (seasonal) hydrological forecasting in the next 3-6 months. While only the ensemble mean (deterministic) forecast skill for streamflow was evaluated in Yuan (2016), both deterministic and probabilistic forecast skill of streamflow (especially for low flows) were assessed in this paper.

3.   **Results**

**3.1 Relation between meteorological drought and hydrological drought**

Figure 5 shows the Pearson correlation coefficients between SPI at different time scales and monthly SSI both for naturalized and observed streamflow. There is an increase in correlation as the SPI time scale increases, suggesting that streamflow is not only influenced by concurrent precipitation, but also by antecedent precipitation up to a few months.

Similar to Vicente-Serrano and López-Moreno (2005), the time scale with the maximum correlation is considered as the time scale of SPI that streamflow responds to. For the naturalized streamflow, it responds to 6-12 months SPI over the upper and middle reaches of Yellow River, and to about 4 months SPI over the lower reaches. The correlations for observed streamflow are significantly lower than for naturalized streamflow for gauges from Xunhua down to Huayuankou, with p values less than 0.01 (Figs. 5b-5j). There is also a significant difference in correlation for Gaocun gauge with p<0.05 (Fig.

5k), but the difference is not statistically significant for Lijin gauge with p>0.1 (Fig. 5l). Except for the Tangnaihai gauge in headwater region, the SPI time scales with the maximum correlation are longer for observed streamflow than that for naturalized streamflow, suggesting that human interventions basically make the hydrological drought respond to longer time scale of meteorological drought. By considering the non-stationarity, the correlations decrease generally from Longmen gauge (Fig. 5h) to the downstream areas, and the decrease is more obvious for observed streamflow and for longer timescale of SPI. This suggests that the relation between meteorological drought and hydrological drought over lower reach of Yellow River is not stationary, anthropogenic climate change and human interventions add more nonlinearity to the propagation from meteorological to hydrological droughts.

In order to analyze the relation during different seasons, five gauges from upper to lower reaches are selected and the correlations between SPI at different time scales and monthly SSI for different target months are plotted in Figure 6. It is found that streamflow responds to shorter time scale of SPI in wet and warm seasons and longer time scale in dry and cold seasons. Again, the correlations for observed streamflow for different target months are lower than that for naturalized streamflow, except for the headwater region (i.e., Tangnaihai gauge) without significant human interventions (Fig. 6). The differences are larger during summer seasons than winter seasons, which is consistent with the seasonality of human water use as shown by different streamflow in Figure 3. Again, non-stationarity basically weakens the relation between meteorological and hydrological droughts over lower reaches, regardless of seasons.

**3.2 Effect of human interventions on the hydrological drought characteristics**

To demonstrate the effect of human interventions on streamflow variations directly, both the time series of naturalized and observed SSIs calculated based on transient climatologies are plotted for the five selected gauges in Figure 7. It is found that human interventions sometimes have positive influence on hydrological drought over the upper reaches. For example, observed SSI can be larger than naturalized SSI at Lanzhou gauge (Fig. 7b). However, those increases mostly occur in winter seasons, while they do decrease in summer season when the water demand and water consumption are high. For the middle and lower reaches, observed SSIs are basically lower than naturalized SSIs (Figs. 7d-e), indicating that human interventions exacerbate hydrological drought conditions in the lower reaches of Yellow River basin. The results based on 50-year climatology generally have lower SSI in 1990s-2000s and higher SSI in 1960s-1970s (not shown), suggesting there are decreasing trends in streamflow over Yellow River basin that is consistent with previous studies (e.g., Piao et al., 2010).

By following the definition of hydrological drought event (it should last for at least 3 continuous months) in Section 2.2, the frequency, duration and severity of hydrological droughts under natural and anthropogenic conditions are calculated and shown in Figure 8. Under natural condition, seasonal hydrological drought occurs 8-16 times during the 50-year (1961-2010) period, where the frequency of hydrological drought is not necessarily higher over lower reaches than that over upper reaches (Fig. 8a). This is partly because there is more precipitation over lower reaches, and the naturalized hydrological drought basically represents the response to the meteorological drought. In contrast, the observed hydrological drought frequency shows quite different characteristics, where the human interventions increase drought frequency by up to 65%

from the upper gauges down to the Xiaheyan gauge (the 5[th] gauge in Fig. 8a), but they increase the drought frequency by 118% or even 262% over the lower reaches (e.g., Lijin gauge at the outlet).

For the drought duration in natural conditions, it is generally longer over upper reaches which is again due to a drier climate (Fig. 8b). With human interventions, there is a slight decrease in drought duration for the upper gauges down to Xiaheyan gauge. This suggests that human interventions reduce the persistency of drought over the upper reaches of Yellow River basin. From Shizuishan gauge (the 6[th] gauge in Fig. 8b) down to the outlet, the duration of hydrological drought increases by 21%-99% under human interventions. There is no significant change in drought severity down to Xiaheyan gauge (the 5[th] gauge in Fig. 8c), but the severity increases by up to 8 times over the lower reaches (Fig. 8c). Therefore, human interventions not only increase drought frequency over most areas of Yellow River basin, but also increase drought severity dramatically over the lower reaches. As compared with the results based on a constant climatology (cyan and pink bars in Fig. 8), the non-stationarity generally reduces human influence on drought frequency, but increases the human impact on drought duration and severity.

To investigate the preference of the occurrence of seasonal hydrological drought, the onset seasons are identified based on individual hydrological drought events. And the ratios of number of drought onsets during different seasons to the total number of drought events are plotted in Figure 9. Without human interventions, most seasonal hydrological droughts start in summer, especially for the upper and lower gauges (Fig. 9a-b). While for the gauges over middle reaches, autumn is also a preferred season for drought onset (Fig. 9a-b). However, Fig. 9 shows that human intervention changes the seasonality of hydrological drought onset significantly, where the spring is the preferred season for the drought onset for the lower reaches, and the ratio for summer season is also increased for the upper reaches. With human interventions, the hydrological drought onset becomes earlier, no matter using transient or constant climatology (Fig. 9c-d).

**3.3 Seasonal forecasting of hydrological drought with human interventions**

To demonstrate the capability of predicting hydrological drought in the anthropocene, a drought case of 2001 is selected to verify the ensemble forecasting of SSI. In terms of meteorological condition, 2001 is a moderate dry year, with precipitation less than the climatology by 9.4%. However, 2001 is a severe hydrological drought year, with observed streamflow less than the climatology by 26%-37% over the upper reaches and by 51%-86% over the lower reaches (http://www.yellowriver.gov.cn/). Figure 10 shows the ensemble forecasts started from February and June of 2001 for the selected five gauges from upper to lower reaches. As verified by the offline simulated SSI, the climatological forecast method (ESP/VIC) has some skill in the February forecast, but totally misses the drought in the June forecast with the ensemble mean SSIs (blue lines in the left panels of Fig. 10) close to zeros that are larger than the drought threshold lines (black horizontal lines) of -0.8. By using the climate-model-based approach (NMME/VIC, see Yuan (2016) for details), there are not significant improvements for the forecasts started from February, but a number of ensembles can capture the hydrological drought conditions for the forecasts started from June, with the ensemble mean SSIs (blue lines in the right panels of Fig. 10) closer to the drought threshold lines. This suggests added values from climate forecast models in the hydrological drought forecasting.

The results shown in Figure 10 neglect the errors or uncertainties in hydrological model because offline simulated hydrological drought index is used as reference to verify the forecasts. To compare with observed SSI, the forecast streamflow series are post-processed (see Section 2.3 for details), and the results for 2001 are shown in Figure 11. Both ESP/VIC and NMME/VIC can predict a drought condition with ensemble mean SSIs much lower than the -0.8 for the middle and lower gauges (Hekouzhen, Huayuankou and Lijin), but both underestimate the drought severity. NMME/VIC shows non-trivial improvement against ESP/VIC in terms of the drought forecasting. An interesting difference is that five gauges have similar ensemble spreads in the natural conditions (Fig. 10), but the spreads vary among upstream and downstream gauges in the anthropogenic conditions (Fig. 11). This is because the hydrological post-processing is applied for each target months and for each gauge independently, the ensemble spreads do not necessarily increase over forecast leads after the post-processing, while they indeed depend on the magnitude or intensity of human interventions.

To assess the probabilistic forecast skill for hydrological droughts during the hindcast period of 1982-2010, the Brier Score (BS; Wilks, 2011) is used. It is defined as:

$$BS = \frac{1}{n}\sum_{k=1}^{n}(y_k - o_k)^2,$$

where n denotes a number of forecast-reference pairs of hydrological drought conditions; $o_k$ is probability from the reference (offline simulated SSI for the natural condition, and observed SSI for the anthropogenic case) for the $k^{th}$ pair, with $o_k$=1 if the drought occurs (SSI<-0.8) and $o_k$=0 if it does not (SSI>-0.8); $y_k$ is the corresponding probability of drought occurrence from the forecast, for example, if 6 of 10 ensemble forecast members have SSI<-0.8, then $y_k$=0.6. BS is similar to the root mean squared error, so a smaller BS represents a better performance.

Figure 12 shows the BS for ensemble hydrological drought forecasts at different hydrological gauges and at different lead times. The statistics are based on a set of comprehensive hindcasts, where for each month during the 29-year (1982-2010) period there is a 6-month hindcast. This consists of 348 hindcast cases, each one is 6-month long, and has 28 or 99 realizations for ESP/VIC and NMME/VIC respectively. For the natural condition (upper panels of Fig. 12), the BS values increase as the forecast leads increase, indicating that the performance generally decreases over leads. The performance of probabilistic hydrological drought forecasting is better over lower reaches than that over upper reaches both for ESP/VIC and NMME/VIC, suggesting the influence of catchment memory. As compared with ESP/VIC, NMME/VIC has a better performance, with BS decreased by 11%-26% in the first month, by 3%-14% in the second and third months.

As verified against observed SSI (lower panels of Fig. 12), the performances for ESP/VIC and NMME/VIC are surprisingly better than the natural cases, and the performances do not necessarily degrade over forecast lead times. This is because human interventions increase the occurrence and severity of hydrological drought, and outweigh the climate variations in many cases. The hydrological post-processing imparts the first-order control in the forecasting, and many post-processed forecasts can represent drought conditions (SSI<-0.8), although they may underestimate the severity as shown in Fig. 11 (right panels). The differences in BS between ESP/VIC and NMME/VIC for the anthropogenic case are smaller than that for

the natural case. In other words, seasonal predictability of hydrological drought in the anthropocene greatly depends on the information of human water use, or the predictability of human interventions.

4.  **Concluding Remarks**

This paper investigates the effects of human interventions on hydrological drought processes and forecasting. Naturalized and observed monthly streamflow are standardized to calculate the hydrological drought index, the Standardized Streamflow Index (SSI). Comparison between naturalized and observed SSI at 12 hydrological gauges along the mainstream of the Yellow River basin (the second largest river basin in China with a drainage area of $7.52×10^5$ km$^2$) shows that human interventions decrease the correlation between hydrological and meteorological droughts, and make the hydrological drought respond to longer time scale of meteorological drought. The relation between meteorological drought and hydrological drought over lower reach of Yellow River is not stationary, anthropogenic climate change and human interventions add more nonlinearity to the propagation from meteorological to hydrological droughts. Seasonal hydrological drought events are identified with monthly SSI<-0.8 for at least three continuous months. Due to heavy human water consumptions over the middle and lower reaches of the Yellow River, there are 118%-262% increases in the drought frequency and up to eight-fold increases in the drought severity, the drought duration increases by 21-99%, and the hydrological drought onset becomes earlier. However, these estimations are heavily based on the quality of naturalized streamflow data. Current procedure of generating naturalized streamflow is basically data driven, where the interaction among different elements is not explicitly considered. In the future, more sophisticated method such as assimilating those precious data into a physical hydrology model that explicitly considers surface water-groundwater interactions and human influences, is necessary for a more robust estimation of naturalized streamflow. Multisource satellite retrieval data (e.g., GRACE terrestrial water storage change, SMAP soil moisture, and MODIS evapotranspiration) could also be a useful complement to in-situ data and hydrological modelling for the estimation.

The naturalized streamflow datasets are used to calibrate the VIC land surface hydrological model and the routing model, and both climatological forcings (ESP) and climate model predicted forcings (NMME) are used to drive the hydrological models to provide seasonal streamflow forecasts. For a severe hydrological drought event occurred over the Yellow River in 2001, ESP/VIC does not capture it while NMME/VIC has some skill when they are verified against naturalized SSI. The added values from climate-model-based seasonal hydrological drought forecasting are decreased in the anthropocene, where both methods can predict a drought condition after the hydrological post-processing but underestimate the severity. Unlike those in naturalized hydrological drought forecasting, the ensemble spreads do not necessarily increase over forecast leads in the anthropocene because of the seasonality of human interventions that have been implicitly incorporated in the hydrological post-processing. Based on the assessment of all hindcasts during 1982-2010, it is found that NMME/VIC decreases (improves) the Brier Score (BS) against ESP/VIC by 11%-26% in the first month and by 3%-14% in the second and third months for the probabilistic hydrological drought forecasting in the naturalized conditions. In the anthropocene, the performances for both forecast methods become better in terms of BS, and the forecast skill does not necessarily decrease over forecast leads due to dominant influence of human water consumption on the hydrological drought processes.

While the effects of human interventions on hydrological drought processes have been studied in the past, to our knowledge, this study is among the first to investigate seasonal hydrological drought forecasting in the anthropocene. Intensive and direct influence of human water use challenges our understanding of hydrological drought predictability. Traditionally, hydro-climate predictability usually refers to the struggle between deterministic and chaotic physical processes. In the

anthropocene, the human influence sometimes outweighs those natural hydro-climate variations and variability, and can be a major source of predictability (or uncertainty if it is not fully understood) for hydrological drought. Current hydrological post-processing procedure accounts for the seasonality of human water use by adding/deducting water to/from the predicted streamflow, and adjusting the forecast results based on the Bayesian theory. Another popular method is to parameterize the human interventions directly in hydrological models (e.g., Wada et al., 2014; Zhou et al., 2016 among others), where the

irrigated area can be estimated for crop water demand, and the diversion and return flows can also be simulated in those models. This modeling framework could be further pushed forward by the availability of Surface Water and Ocean Topography (SWOT) satellite data in the near future, where surface water storage in reservoirs and rivers can be monitored at 250m resolution (Biancamaria et al., 2016). However, it is difficult to account for the inter-annual variability of human water use in a "real" forecasting mode. During severe hydrological drought events, the human water use tends to be more

intensive to adapt to drought conditions. How to quantify and model the variability of human water use is an interdisciplinary question for both physical and social sciences. In addition, the interdisciplinary collaboration is also indispensable to objectively quantify and to accurately predict drought impact, as the drought impact and drought are quite different.

**Acknowledgement.** We would like to thank Prof. Dennis Lettenmaier and three anonymous reviewers for their helpful comments. This work was supported by the National Natural Science Foundation of China (No. 91547103), China Special Fund for Meteorological Research in the Public Interest (Major projects) (GYHY201506001), and the Thousand Talents Program for Distinguished Young Scholars. This paper is submitted to the special issue "Observations and modeling of land surface water and energy exchanges across scales" in honor of Professor Eric F. Wood. The first author would like to thank

Eric for his inspiring guidance.

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

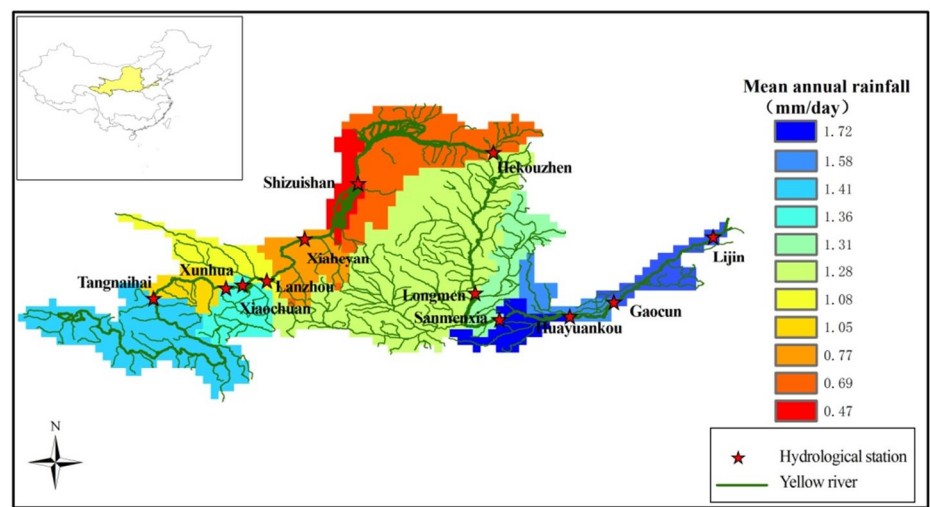

**Figure 1.** Locations of hydrological stations over Yellow River basin. Shaded areas are regional mean annual rainfall (mm/day) averaged during 1961-2010.

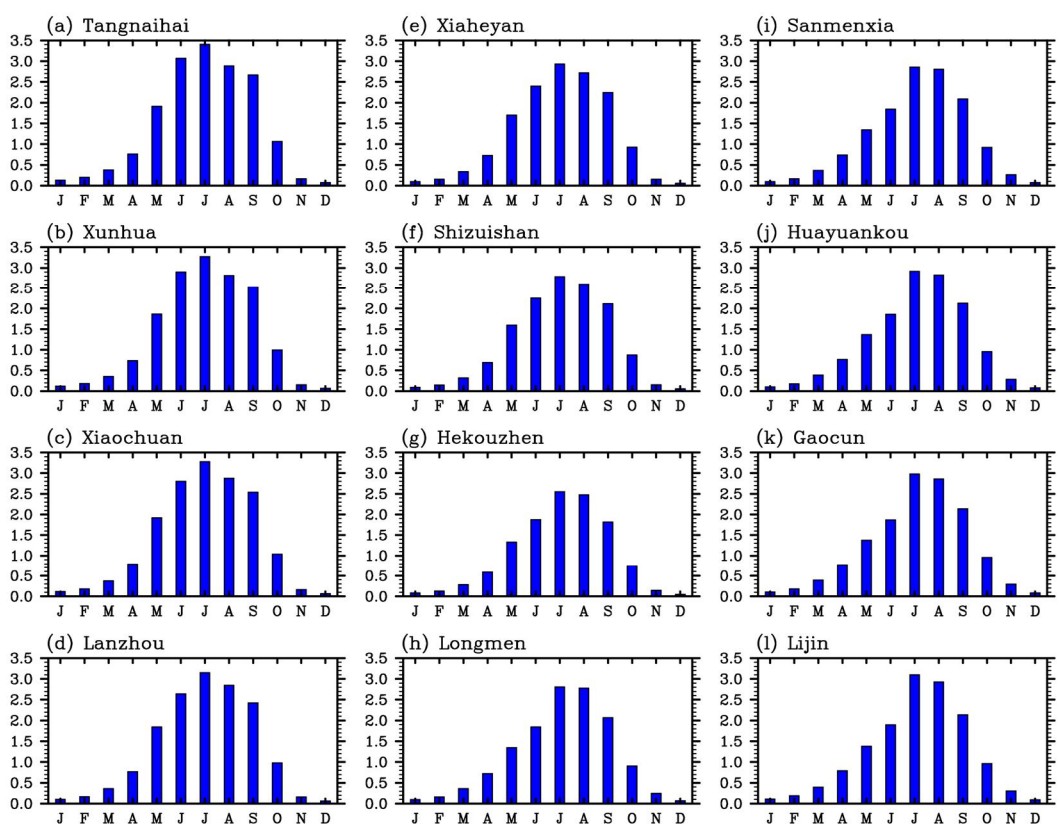

**Figure 2.** Monthly mean rainfall (mm/day) averaged over 1961-2010 for each sub-basin.

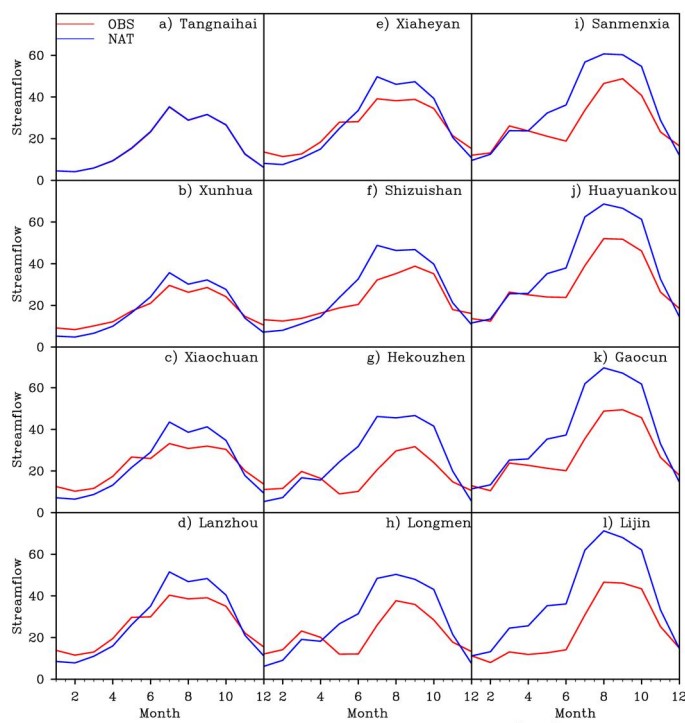

**Figure 3.** Monthly mean naturalized (blue) and observed (red) streamflow ($10^8$ m$^3$) averaged over 1961-2010 for 12 hydrological gauges located from upper to lower mainstream of Yellow River.

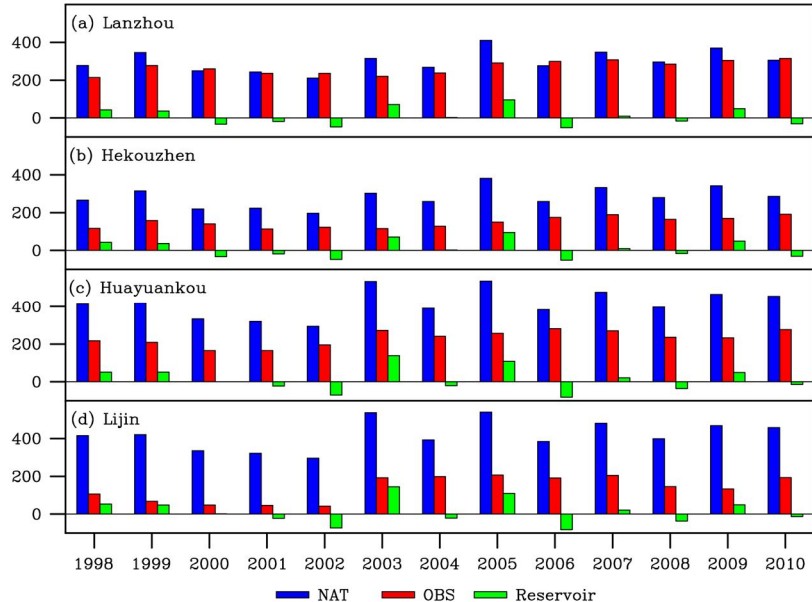

5   **Figure 4.** Annual mean naturalized (blue) and observed (red) streamflow ($10^8$ m$^3$), and reservoir storage change ($10^8$ m$^3$, negative green values represent reservoir water distribution) accumulated within four selected sub-basins (from headwater down to the gauge) during 1998-2010.

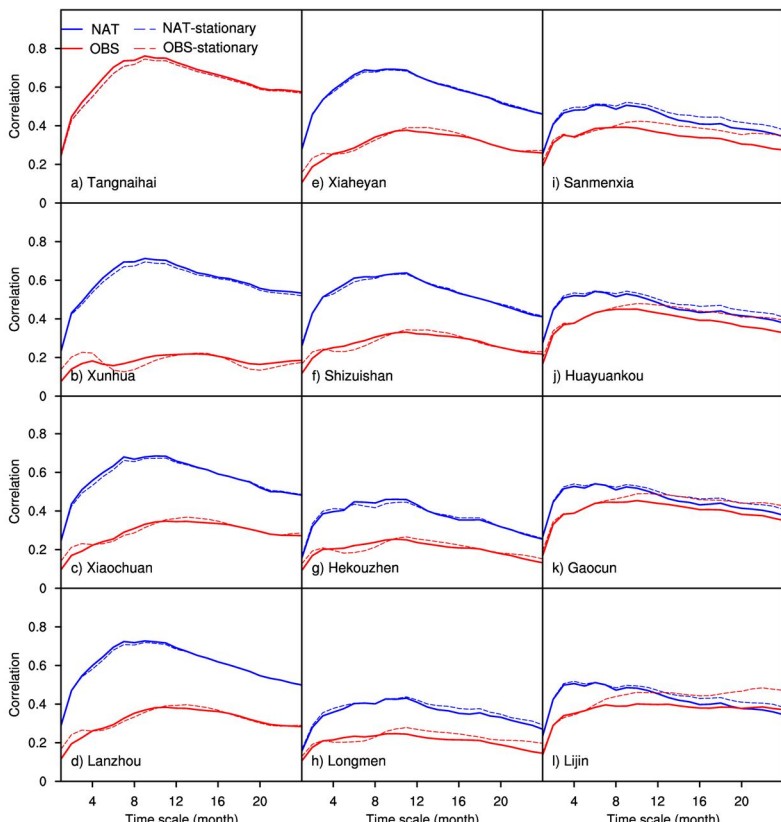

**Figure 5.** Correlations between Standardized Precipitation Index (SPI) at different time scales and monthly naturalized (blue) or observed (red) Standardized Streamflow Index (SSI) for 12 hydrological gauges located from upper to lower mainstream of Yellow River. The solid lines are for SPI and SSI calculated based on transient 30-year climatologies with consideration of non-stationarity, while the dashed lines are for those based on the 50-year (1961-2010) climatology.

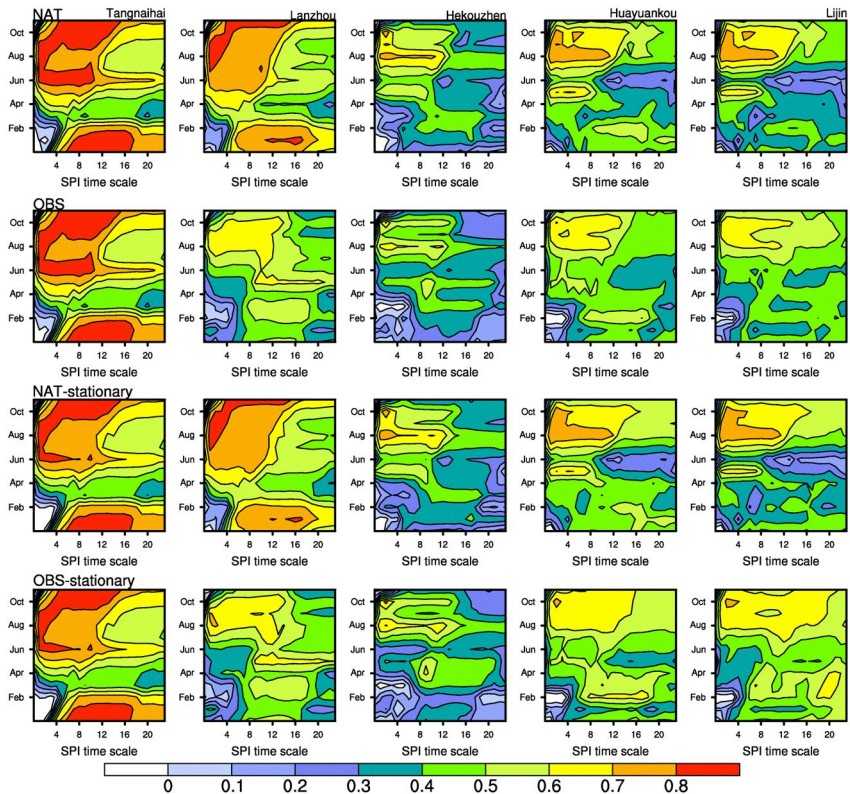

**Figure 6.** The same as Fig. 5, but for each target month for five selected hydrological gauges.

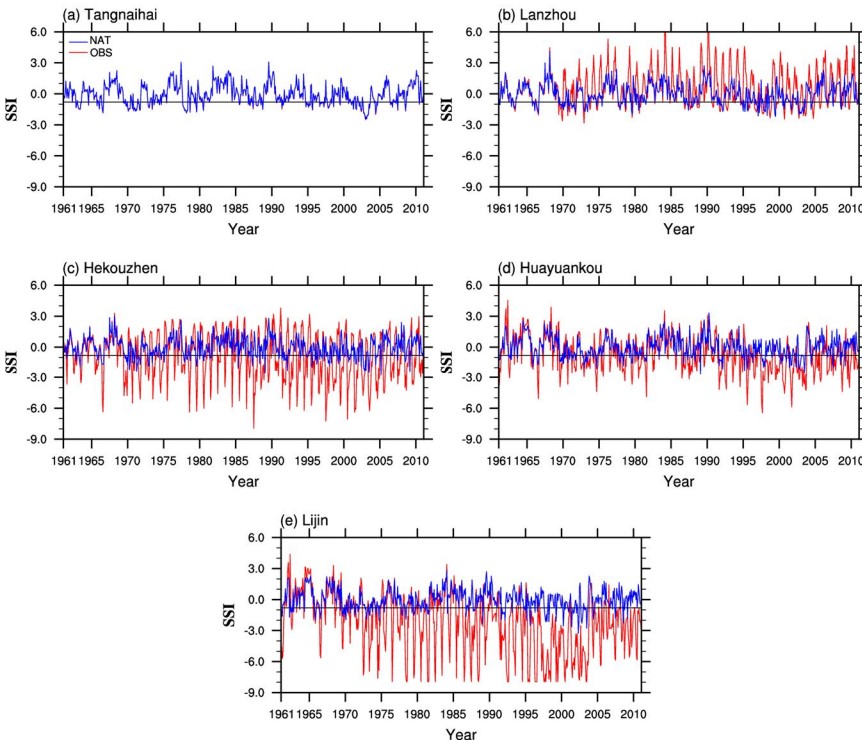

**Figure 7.** Time series of naturalized (blue) and observed (red) 1-month Standardized Streamflow Index (SSI) for five selected hydrological gauges. SSI is calculated based on transient 30-year climatologies. The horizontal black lines represent the threshold of -0.8 for drought conditions.

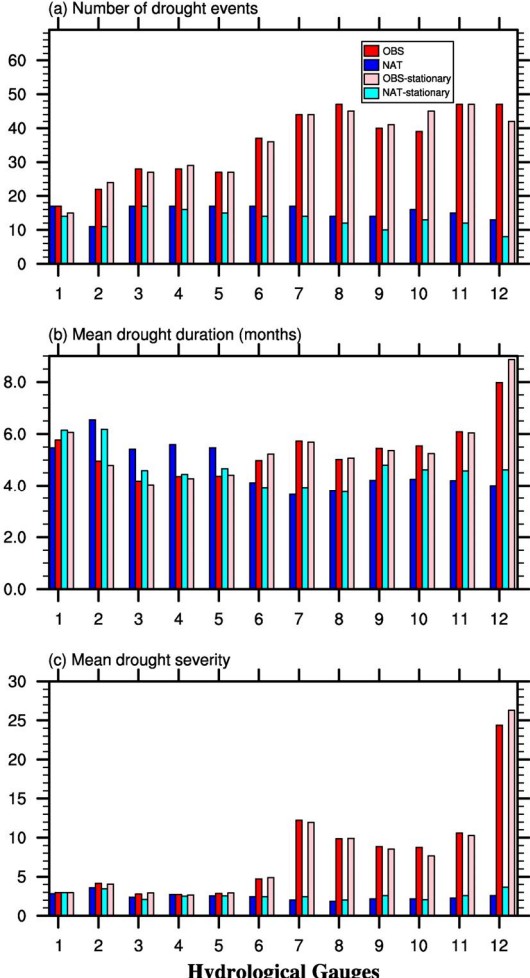

**Figure 8.** Characteristics of hydrological drought events based on streamflow time series with or without human influences during 1961-2010 for 12 hydrological gauges. A hydrological drought event is selected when the Standardized Streamflow Index (SSI) is continuously below -0.8 for at least 3 months. Blue and red bars are for the results based on transient 30-year climatologies, and cyan and pink bars are for results based on the 50-year (1961-2010) climatology.

**Seasonal onset ratio of hydrological drought**

**Figure 9.** Ratios of the number of hydrological drought onsets occurring in different seasons to the total number of hydrological drought events for the 12 hydrological gauges during 1961-2010. Drought events are classified the same as those in Fig. 8. The left for the results based on transient 30-year climatologies, and the right are for results based on the 50-year (1961-2010) climatology.

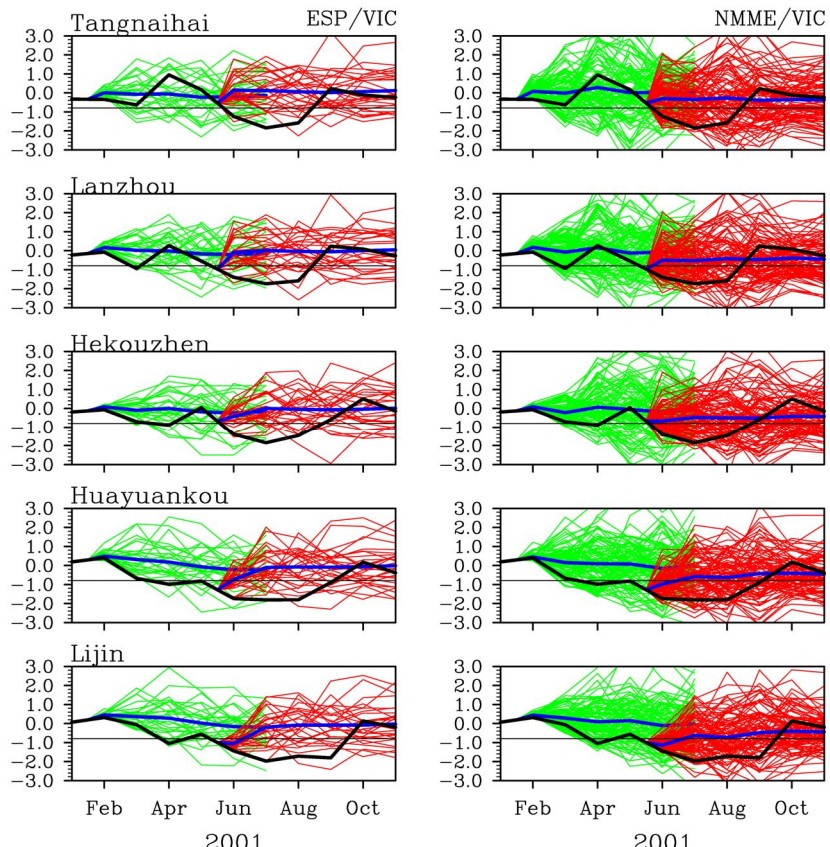

**Figure 10.** Seasonal ensemble hindcast of the 2001 Yellow River hydrological drought from upstream to downstream gauges by using a climatology method (ESP/VIC) and the climate-model-based approach (NMME/VIC). Vertical axes are Standardized Streamflow Index (SSI), where SSI<-0.8 represents a hydrological drought condition. Solid black lines represent the offline simulated SSI by the hydrological model VIC, green and red lines are for individual ensemble members from the hindcasts started from the beginnings of February and June respectively, and blue lines are the ensemble means of the hincasts.

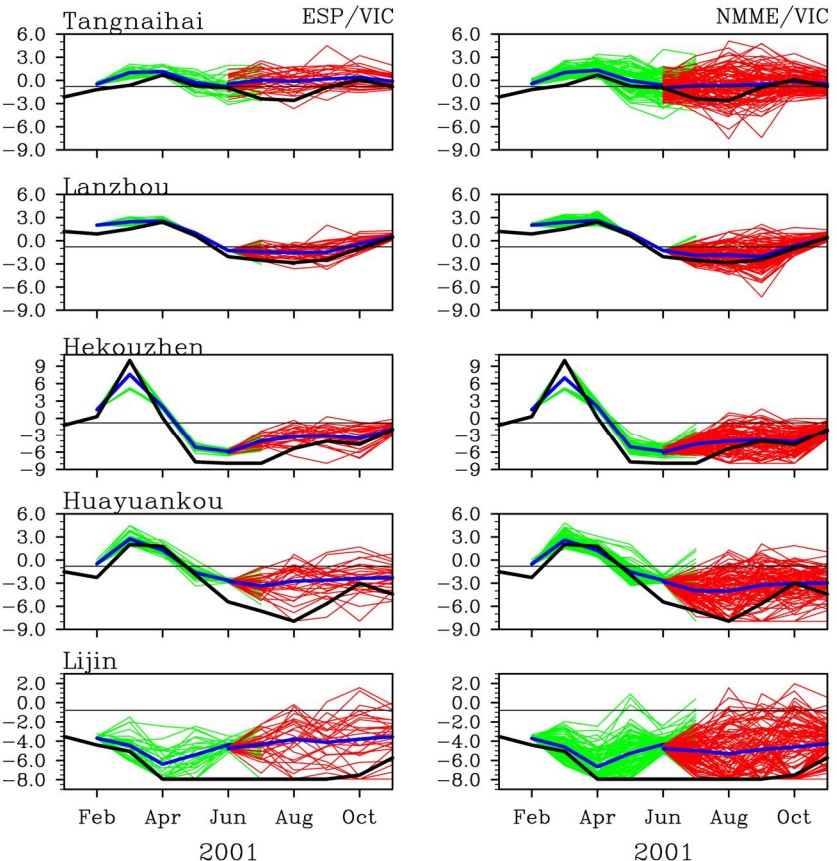

**Figure 11.** The same as Fig. 10, but for the post-processed Standardized Streamflow Index (SSI) hindcasts from ESP/VIC and NMME/VIC (see Section 2.3 for details) as verified by observed SSI.

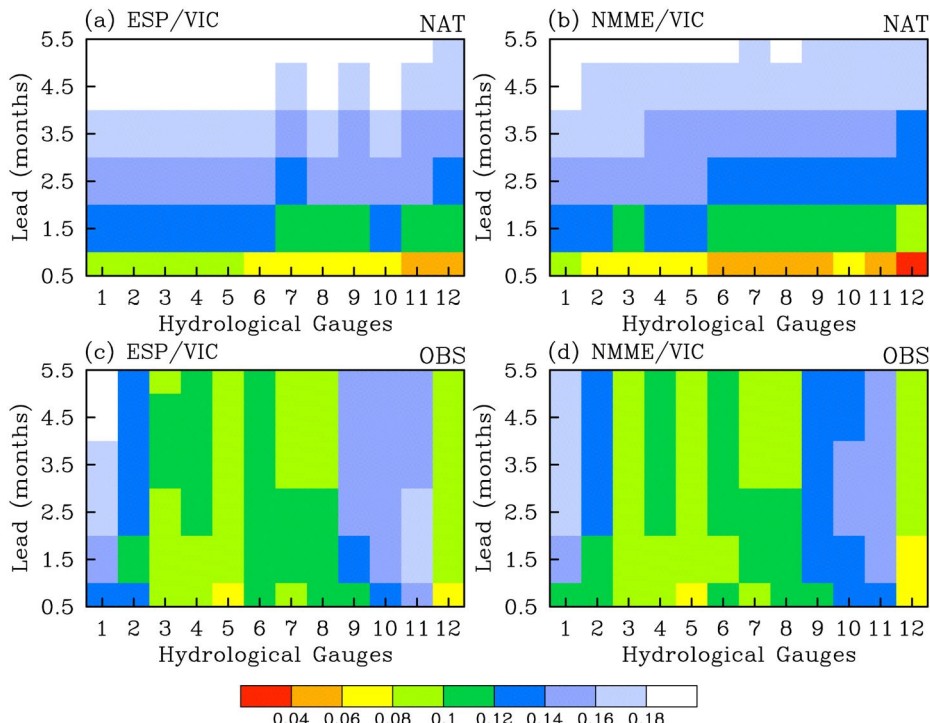

**Figure 12.** Brier score (BS) for ensemble hydrological drought forecasts at different lead times from a climatology method (ESP/VIC) and the climate-model-based approach (NMME/VIC) for different hydrological gauges as verified against VIC offline simulated (a, b) and observed (c, d) streamflow over Yellow River basin during the period of 1982-2010. For the verification against observed streamflow, the forecasts have been post-processed.