# Peer review of "Understanding and seasonal forecasting of hydrological drought in the anthropocene"

_Hydrology and Earth System Sciences, 2016_

## Referee Comment (RC1) · Anonymous Referee #1 · 7 Dec 2016

This paper examines hydrological drought in managed river basins in China. Based on the 29-year NMME forecasts, they compared the skill of hydrological drought forecasts between the naturalized and observed conditions . They found that human intervention out weighted the climate variability for hydrological drought forecasts. The paper is well written. I recommend to be published after minor revisions. My specific comments are listed below:

1. Statistical significance : This may be the weakness of this paper. You compared the correlation between naturalized and observed SPI or SSI. There is no statistical assessment. For example: Fig2. Some gauge like Xunhua, the differences are significant, but differences for gauge Lijin may be not. You need to add statistical significance test to results. 2. Is SSI similar to standardized runoff index (Shukla and Wood 2008) except you use streamflow? 3. Section 2.3 Please add more details For the VIC simulation, what are the sources for daily precipitation and temperature time series used to derive forcings? Did you run the VIC model for these 12 gauge sites or the whole domain? Did you use the VIC in the water balance mode ( no observed radiation terms) ? which version? 4. Drought is usually defined as persistent low flow conditions. Does naturalized drought persist longer? Please comment on the persistence of low flow (SSI) conditions. 5. You used NMME forecasts. Did you perform hydroclimate forecasts using VIC for each model separately and then took the ensemble means? How exactly did you process the NMME data? Readers need more details on that 6. You stated seasonal cycle plays a role in drought. (page 5 response time is different for summer and winter). How large is the precipitation seasonal cycle?

---

## Referee Comment (RC2) · Anonymous Referee #2 · 12 Dec 2016

Overall Remarks

The paper presents a fairly interesting study on an important topic with substantial results and insights. The research therein is a good fit for HESS. The main focus is on the impact of human water use/regulation activities on drought. The authors also carried out a number of seasonal meteorological/hydrological forecast experiments and I find them very carefully designed and carried out. The results/discussions are clearly presented too. My major concerns are on the analysis methodology and the adequacy of supporting information. The study area is one large river basin in China while a quite minimum level of specific information on the local water management is provided. Usually, more information on the surface water use practices will be very useful in helping readers understand the findings and their implications across similar areas in other parts of the world. I recommend its publication in HESS with improvements on

the analysis method and additional discussion on local water management and how that leads to what is seen in the results.

Specific Remarks

The paper (circa P. 5, L. 3-18) interprets the peak correlation time scale as the "optimal response time of streamflow to sub-basin averaged precipitation", while offering no supporting evidence (e.g. citation of previous research, data results). The 6-12 months (and later 8-16 months) "response time" seems incredibly long and beyond what a hydrologist can reasonably expect. Given the size of the Yellow River basin, it shouldn't take more than a month or two for water to travel from rain-falling hillslopes down to river gauging stations. And the local soil water stores or snowpack won't be able to defer the release of precipitated water for that long either. SPI/SSI does time averaging to the underlying parameters and this essentially smooths out noises at shorter time scales. A true "response time" is usually calculated from time lagged correlation analysis, e.g., between SPI-1 and SSI-1. Either the "response time" needs to be calculated differently or the same calculations need to be interpreted differently. Note that the change in the relationship between meteorology and hydrology is one of the major points in the paper as summarized in the abstract.

Further, the notion of "nonlinear response" of hydrological drought to meteorological drought is a bit vague in the discussions. The rainfall-runoff process is by itself "non-linear" and lagged in time, at least at short time scales. If the word "response" refers to the rainfall-runoff process (at any time scale), the research here should try to find out what exactly human interventions did to that process. Reduction in streamflow volume (i.e. significant amount of consumptive use)? Longer lag times (delayed release for flood control)? If the lag times become longer, should this be considered in the fore-cast post-processing procedure? (For example, a time series based procedure that looks at a prior history instead of just the current month.) For the same reasons, more information on the water regulation practices in the study area is needed for a (much) better understanding of the impacts and differences found in the results. For example, reservoirs may store rain water from wet season and distribute it in the remaining time of the year according to the need. How much of the streamflow water is being regulated in the Yellow River basin (e.g. reservoir capacity relative to the annual total inflow) and for what purposes? How much of the streamflow is being modified (in both absolute and relative senses)? Fig. 4 helps to understand the scenario but direct comparisons between observations and naturalized values (in seasonal cycles and annual totals) can help explain what happened in Fig. 4 in a much better way. I guess the observed SSI in Fig. 4 is calculated against observed flow climatology and naturalized SSI against naturalized flow, right? (Please clarify.) If so, the comparisons between the two do not reveal the difference between the observed and naturalized climatologies, e.g. reduced total flow volumes or lagged peak times. Specific information on the local water management and water use practices is always helpful in understanding the findings and their implications across similar areas in other parts of the world (Wada et al., 2014). The study could be significantly stronger if more specific water management information is provided and related to the research findings.

P. 5, L. 13: nonlinearly -> nonlinear Fig. 1: The map needs to show at least the Yellow River and its main tributaries (thicker lines for the main stream) under this study. Replace the political boundaries with sub-basin boundaries (keep the coast lines). Fig. 4: SSI at what time scale? 1-month? Subplots are too small and better if they are rearranged into multiple columns.

References

Wada, Y., Wisser, D., and Bierkens, M. F. P.: Global modeling of withdrawal, allocation and consumptive use of surface water and groundwater resources, Earth Syst. Dynam., 5, 15-40, doi:10.5194/esd-5-15-2014, 2014.

---

## Author Comment (AC1) · 7 Jan 2017

**Responses to the comments from Reviewer #1**

We are very grateful to the Reviewer for the positive and careful review. The thoughtful comments have helped improve the manuscript. The reviewer's comments are italicized and our responses immediately follow.

*This paper examines hydrological drought in managed river basins in China. Based on the 29-year NMME forecasts, they compared the skill of hydrological drought forecasts between the naturalized and observed conditions. They found that human intervention out weighted the climate variability for hydrological drought forecasts. The paper is well written. I recommend to be published after minor revisions. My specific comments are listed below.*

**Response:** We would like to thank the reviewer for the positive comments. Please see our responses below.

*Statistical significance: This may be the weakness of this paper. You compared the correlation between naturalized and observed SPI or SSI. There is no statistical assessment. For example: Fig2. Some gauge like Xunhua, the differences are significant, but differences for gauge Lijin may be not. You need to add statistical significance test to results.*

**Response:** Thanks for the comments. We have incorporated the statistical significance testing and revised the manuscript as follows, where Fig. 2 is Fig. 5 now due new addition of three figure before it.

"The correlations for observed streamflow are significantly lower than for naturalized streamflow for gauges from Xunhua down to Huayuankou, with p values less than 0.01 (Figs. 5b-5j). There is also a significant difference in correlation for Gaocun gauge with $p<0.05$ (Fig. 5k), but the difference is not statistically significant at 90% level for Lijin gauge with $p>0.1$ (Fig. 5l)."

*Is SSI similar to standardized runoff index (Shukla and Wood 2008) except you use streamflow?*

**Response:** Exactly. We have clarified it in the revised manuscript as follows:

" Note that SSI was similar to the standardized runoff index (SRI) defined by Shukla and Wood (2008), except that streamflow was used here for a standardization."

*Section 2.3 Please add more details For the VIC simulation, what are the sources for daily precipitation and temperature time series used to derive forcings? Did you run the VIC model for these 12 gauge sites or the whole domain? Did you use the VIC in the water balance mode ( no observed radiation terms)? which version?*

**Response:** Thanks for the comment. We have clarified the information for the VIC simulation as follows:

"The observed meteorological forcing datasets including daily precipitation, daily maximum and minimum surface air temperature, and surface wind were interpolated from 324 China

Meteorological Administration stations. And the VIC model version 4.0.5 was used to predict runoff in a water balance mode over the entire Yellow River basin with 1321 grid cells at 0.25-degree resolution (Yuan et al., 2016)."

*Drought is usually defined as persistent low flow conditions. Does naturalized drought persist longer? Please comment on the persistence of low flow (SSI) conditions.*

**Response:** In this study, the monthly streamflow records were converted into percentiles to represent the low flow conditions at seasonal time scale. A hydrological drought event was defined as follows:

"A threshold of -0.8 was used to represent a drought condition for both SPI and SSI. And a hydrological drought event was selected when the SSI was below -0.8 for at least 3 continuous months (Yuan and Wood, 2013)…"

As shown in Figure 5b (Figure 8b in the revised manuscript), the naturalized drought persists a little longer than the observed streamflow at upper gauges, suggesting the positive influence of human intervention. However, the former is basically shorter than the latter at middle and lower gauges, mainly due to intensive human water consumption. We clarified the duration changes in the manuscript as follows:

" For the drought duration in natural conditions, it is generally longer over upper reaches which is again due to a drier climate (Fig. 8b). With human interventions, there is a decrease in drought duration for the upper gauges down to Xiaheyan gauge. This suggests that seasonality of human interventions reduces the persistency of drought over the upper reaches of the Yellow River basin. From Shizuishan gauge (the 6$^{th}$ gauge in Fig. 8b) down to the outlet, the duration of hydrological drought increases by 12%-83% under human interventions."

*You used NMME forecasts. Did you perform hydroclimate forecasts using VIC for each model separately and then took the ensemble means? How exactly did you process the NMME data? Readers need more details on that.*

**Response:** Yes, the VIC model was driven by each ensemble members of nine NMME models with a grand ensemble of 99 realizations, and ensemble means were then calculated to evaluate the deterministic forecast skill. In addition, all 99 realizations were used directly to evaluate the probabilistic forecast skill.

The downscaling process is the same as Yuan (2016). To have this paper self-contained, we have added some descriptions as follows:

"Thirdly, a grand ensemble of 99 realizations from eight North American Multimodel Ensemble (NMME; Kirtman et al., 2014) models was used to force hydrological models to generate the NMME/VIC hindcast dataset (Yuan, 2016). Here, the 1-degree NMME global hindcasts of monthly precipitation and temperature were bilinearly interpolated into 0.25-degree, the interpolated monthly hindcasts for each calendar month and each NMME model were then bias-corrected independently against observations by using the quantile-mapping method (Wood et al., 2002) in a cross-validation mode (i.e., dropping observation and forecast in the target year when

building the climatology), and these bias-corrected monthly hindcasts were finally temporally disaggregated to daily by historical sampling and rescaling (Yuan, 2016)."

*You stated seasonal cycle plays a role in drought. (page 5 response time is different for summer and winter). How large is the precipitation seasonal cycle?*

**Response:** Thanks for the comment, we have added a figure to show the precipitation seasonal cycle (Figure 2 in the revised manuscript), and have clarified the seasonal cycle as follows:

"Most precipitation over Yellow River occurs in summer season due to the influence of East Asian monsoon, resulting in a strong seasonality of precipitation, with more than 80% of annual precipitation falls within May-September (Fig. 2)."

[Figure]

**Figure 2.** Monthly mean rainfall (mm/day) averaged over 1961-2010 for each sub-basin.

**References:**

Shukla, S., and Wood, A. W.: Use of a standardized runoff index for characterizing hydrologic drought, Geophys. Res. Lett., 35, L02405, doi:10.1029/2007GL032487, 2008.

Wood, A. W., Mauer, E. P., Kumar, A., and Lettenmaier, D. P.: Long-range experimental hydrologic forecasting for the eastern United States, J. Geophys. Res., 107, 4429, doi:10.1029/2001JD000659, 2002.

Yuan, X.: An experimental seasonal hydrological forecasting system over the Yellow River basin – Part 2: The added value from climate forecast models, Hydrol. Earth Syst. Sci., 20, 2453–2466, doi:10.5194/hess-20-2453-2016, 2016.

---

## Author Comment (AC2) · 7 Jan 2017

**Responses to the comments from Reviewer #2**

We are very grateful to the Reviewer for the positive and careful review. The thoughtful comments have helped improve the manuscript. The reviewer's comments are italicized and our responses immediately follow.

*The paper presents a fairly interesting study on an important topic with substantial results and insights. The research therein is a good fit for HESS. The main focus is on the impact of human water use/regulation activities on drought. The authors also carried out a number of seasonal meteorological/hydrological forecast experiments and I find them very carefully designed and carried out. The results/discussions are clearly presented too. My major concerns are on the analysis methodology and the adequacy of supporting information. The study area is one large river basin in China while a quite minimum level of specific information on the local water management is provided. Usually, more information on the surface water use practices will be very useful in helping readers understand the findings and their implications across similar areas in other parts of the world. I recommend its publication in HESS with improvements on the analysis method and additional discussion on local water management and how that leads to what is seen in the results.*

**Response:** We would like to thank the reviewer for the positive comments. We have added two new figures and the corresponding text to provide the water management information in details, and have revised the interpretation of the relation between meteorological and hydrological drought. Please see our responses below.

*Specific Remarks*

*The paper (circa P. 5, L. 3-18) interprets the peak correlation time scale as the "optimal response time of streamflow to sub-basin averaged precipitation", while offering no supporting evidence (e.g. citation of previous research, data results). The 6-12 months (and later 8-16 months) "response time" seems incredibly long and beyond what a hydrologist can reasonably expect. Given the size of the Yellow River basin, it shouldn't take more than a month or two for water to travel from rain-falling hillslopes down to river gauging stations. And the local soil water stores or snowpack won't be able to defer the release of precipitated water for that long either. SPI/SSI does time averaging to the underlying parameters and this essentially smooths out noises at shorter time scales. A true "response time" is usually calculated from time lagged correlation analysis, e.g., between SPI-1 and SSI-1. Either the "response time" needs to be calculated differently or the same calculations need to be interpreted differently. Note that the change in the relationship between meteorology and hydrology is one of the major points in the paper as summarized in the abstract.*

**Response:** We greatly appreciate the positive comment. We calculated the response time as suggested by the reviewer and found that the most significant lag correlations occur at lag-1

month both for naturalized and observed streamflow along the mainstream of Yellow River, although the lag-correlations are again lower for the observed streamflow.

Actually in the last version of the manuscript, we followed the work done by Vicente-Serrano and López-Moreno (2005), and thought the time scale with the maximum correlation is considered as the time scale of SPI that streamflow responds to, i.e., the SPI time scale that has the most similar variations to the SSI. However, we have realized using "response time" in the manuscript would be very confusing. So, we have removed all "response time" throughout the paper, and have re-written the corresponding text as follows:

**Abstract**—"It is found that human interventions decrease the correlation between hydrological and meteorological droughts, and make the hydrological drought respond to longer time scale of meteorological drought especially during rainy seasons."

**Section 3.1**—"Similar to Vicente-Serrano and López-Moreno (2005), the time scale with the maximum correlation is considered as the time scale of SPI that streamflow responds to. For the naturalized streamflow, it responds to 6-12 months SPI over the upper and middle reaches of Yellow River, and about 4 months SPI over the lower reaches … Except for the Tangnaihai gauge in headwater region, the SPI time scales with the maximum correlation are longer for the observed streamflow, suggesting that human interventions basically make the hydrological drought respond to longer time scale of meteorological drought … It is found that streamflow responds to shorter time scale of SPI in wet and warm seasons and longer time scale in dry and cold seasons. "

**Concluding Remarks**—"Comparison between naturalized and observed SSI at 12 hydrological gauges along the mainstream of the Yellow River basin (the second largest river basin in China with a drainage area of $7.52 \times 105$ km2) shows that human interventions decrease the correlation between hydrological and meteorological droughts, and make the hydrological drought respond to longer time scale of meteorological drought especially during rainy seasons."

*Further, the notion of "nonlinear response" of hydrological drought to meteorological drought is a bit vague in the discussions. The rainfall-runoff process is by itself "nonlinear" and lagged in time, at least at short time scales. If the word "response" refers to the rainfall-runoff process (at any time scale), the research here should try to find out what exactly human interventions did to that process. Reduction in streamflow volume (i.e. significant amount of consumptive use)? Longer lag times (delayed release for flood control)? If the lag times become longer, should this be considered in the forecast post-processing procedure? (For example, a time series based procedure that looks at a prior history instead of just the current month.)*

**Response:** We have also removed "nonlinear response" in the revised manuscript given that the focus of this work is not to investigate the lag-correlation for forecasting. We have plotted the annual cycle of naturalized and observed streamflow in Figure 3 to show the effect of human interventions, and revised the section "2.1 study domain and hydroclimate observation data" as follows:

"In general, human interventions decrease streamflow over upper and middle reaches of Yellow River during rainy season while increase it during dry season (Figs. 3b-3h). This suggests that reservoirs in the upper and middle reaches store rain water in wet season and distribute it in the remaining time of the year according to the need, which is similar to regulations in other parts of the world (Wada et al., 2014). Actually, Figure 4a shows that the annual mean observed streamflow at upper reaches can be higher than the naturalized streamflow during dry years due to the reservoir water release (e.g., years 2000, 2002, 2006 and 2010 for Lanzhou gauge). Over the lower reaches, the observed streamflow is significantly lower than the naturalized streamflow during wet season due to heavy water consumption (Figs. 3i-3l), but the former is close to the latter during dry season because of no significant water consumption or reservoir management."

[Figure]

**Figure 3.** Monthly mean naturalized (blue) and observed (red) streamflow ($10^8$ m$^3$) averaged over 1961-2010 for 12 hydrological gauges located from upper to lower mainstream of the Yellow River.

*For the same reasons, more information on the water regulation practices in the study area is needed for a (much) better understanding of the impacts and differences found in the results. For example, reservoirs may store rain water from wet season and distribute it in the remaining time of the year according to the need. How much of the streamflow water is being regulated in the Yellow River basin (e.g. reservoir capacity relative to the annual total inflow) and for what purposes? How much of the streamflow is being modified (in both absolute and relative senses)?*
**Response:** We thank for the comment. We have now collected annual statistics for the reservoir storage change during 1998-2010, but failed to obtain the monthly data. Based on the data

available, we have added Figure 4 to show the interannual variations of naturalized and observed streamflow and the reservoir storage change, and we have revised the manuscript as follows:

"Figure 4 also shows that the magnitudes of reservoir storage changes are quite small as compared with streamflow. In fact, the mean absolute changes of reservoir storage during 1998-2010 are about 14%-38% and 12%-14% of observed and naturalized streamflow, respectively. This suggests that other human interventions, such as direct withdrawal of surface water for agricultural, industrial and civil consumptions, account for a large part of streamflow variations over Yellow River."

[Figure]

**Figure 4.** Annual mean naturalized (blue) and observed (red) streamflow ($10^8$ m$^3$), and reservoir storage change ($10^8$ m$^3$, negative green values represent reservoir water distribution) within four selected sub-basins during 1998-2010.

*Fig. 4 helps to understand the scenario but direct comparisons between observations and naturalized values (in seasonal cycles and annual totals) can help explain what happened in Fig. 4 in a much better way. I guess the observed SSI in Fig. 4 is calculated against observed flow climatology and naturalized SSI against naturalized flow, right? (Please clarify.) If so, the comparisons between the two do not reveal the difference between the observed and naturalized climatologies, e.g. reduced total flow volumes or lagged peak times.*

**Response:** Observed SSI in Fig.4 is not calculated against observed flow climatology. Actually both naturalized and observed SSI are calculated against the naturalized flow climatology, so they can be compared to detect the effect of human interventions on hydrological drought. Seasonal cycle of original values are now shown in Figure 3 (please see our response above) to support the SSI analysis.

*Specific information on the local water management and water use practices is always helpful in understanding the findings and their implications across similar areas in other parts of the world (Wada et al., 2014). The study could be significantly stronger if more specific water management information is provided and related to the research findings.*

**Response:** Thanks for the comment. Two figures regarding the seasonal cycle of monthly naturalized and observed streamflow, and the annual mean streamflow and reservoir storage change have been added into the revised manuscript. Please see our responses above.

*P. 5, L. 13: nonlinearly -> nonlinear*

**Response:** Revised as suggested.

*Fig. 1: The map needs to show at least the Yellow River and its main tributaries (thicker lines for the main stream) under this study. Replace the political boundaries with sub-basin boundaries (keep the coast lines).*

**Response:** We have revised Figure 1 as suggested.

[Figure]

**Figure 1.** Locations of hydrological stations over the Yellow River basin. Shaded areas are regional mean annual rainfall (mm/day) averaged during 1961-2010.

*Fig. 4: SSI at what time scale? 1-month? Subplots are too small and better if they are rearranged into multiple columns.*

**Response:** Fig. 4 (Fig. 7 in the revised manuscript) has been replotted to show the panels in two columns. The SSI is at 1-month time scale, and it has been clarified in the revised figure caption:

[Figure]

**Figure 7.** Time series of naturalized (blue) and observed (red) 1-month Standardized Streamflow Index (SSI) for five selected hydrological gauges. The horizontal black lines represent the threshold of -0.8 for drought conditions.

**References:**

Vicente-Serrano, S. M. and López-Moreno, J. I.: Hydrological response to different time scales of climatological drought: an evaluation of the Standardized Precipitation Index in a mountainous Mediterranean basin, Hydrol. Earth Syst. Sci., 9, 523–533, doi:10.5194/hess-9-523-2005, 2005.

Wada, Y., Wisser, D., and Bierkens, M. F. P.: Global modeling of withdrawal, allocation and consumptive use of surface water and groundwater resources, Earth Syst. Dynam., 5, 15-40, doi:10.5194/esd-5-15-2014, 2014.

---

## Author Response (AR1)

[revised manuscript text omitted]

Email: yuanxing@tea.ac.cn
Tel: +86-10-82995385
http://www.escience.cn/people/yuanxing

March 22, 2017

Prof. Dennis Lettenmaier
Special Issue Editor
Hydrology and Earth System Sciences

RE: manuscript #hess-2016-592

Dear Prof. Lettenmaier,

Thank you for your kind decision letter on our manuscript entitled "Understanding and seasonal forecasting of hydrological drought in the anthropocene" (hess-2016-592). We have carefully considered your and reviewer's comments and incorporated them into the revised manuscript to the extent possible. Major changes include adding three new figures and the corresponding text to provide the climate and water management information in details, revising the interpretation of the relation between meteorological and hydrological drought, and clarifying the moviations and methods. We hope that you find the revised manuscript and the response to the reviews acceptable to *HESS*.
The detailed responses to the comments are attached.

We appreciate the effort you spent to process the manuscript and look forward to hearing from you soon.

Sincerely yours,

Xing Yuan

**Responses to the comments from Editor**

We are very grateful to the Editor for the positive and careful review. The thoughtful comments have helped improve the manuscript. The editor's comments are italicized and our responses immediately follow.

5 *This paper analyzes precipitation and streamflow in the Yellow River basin using the standardized precipitation and streamflow indices (basically just Z-values, where the percentiles come from a fitted gamma distribution). They analyze both naturalized flows (it's not entirely clear where they came from, but probably an attempt by someone to back out the effects of irrigation and reservoir storage) and observed flows. As is well known, irrigation development in the basin has increased rapidly in recent*
10 *decades, and during the irrigation season flows have been greatly diminished in some reaches. This shows up in their Figure 4, where for basins like Lijin in particular, the observed flows have SSI values that go as low as -9, which would be an absurdly small value for the naturalized flows (for which something in excess of 99% of the values should be between +/- 3).*

**Response:** Thanks for the comments. As explained in the manuscript "… the naturalized streamflow
15 was calculated by adding the water consumed by agricultural, industrial and civil sectors, and the water regulated by reservoirs, back to the observed streamflow (Yuan et al., 2016)." (P4L1-3 in this response letter)

Yuan et al. (2016) used the naturalized streamflow to calibrate the VIC model without water management model during 1961-1981, when the human interventions were supposed to be limited. The
20 naturalized streamflow was then compared with VIC simulations during 1982-2010, and their NSEs vary between 0.71-0.91. And as shown in Figure S1 below, the naturalized streamflow agrees with VIC simulations quite well. In addition, we also compared the naturalized streamflow with another version of VIC model simulation (Zhou et al., 2016) during 1961-2010, and resulted in NSEs vary between 0.53-0.71 (Fig. S2). So, we believe the naturalized streamflow is a reliable source of data that supports
25 the investigation of human impacts as compared with observed streamflow in this study.

For the low SSI values at Lijin gauge, we have checked the programs and scripts and find that those low SSI values (around -8) are associated with very low streamflow or even zero streamflow. Moreover, the observed SSI was calculated by using the distribution parameters fitted from naturalized streamflow instead of the observed streamflow, this also increases the chance of extremely low SSI values. As
30 shown in Figure S3, the observed streamflow is significantly lower than the naturalized streamflow during 1981-2010. We have also tested the SSI by using observed streamflow to fit gamma distributions, and as commented by the editor, most values are between +/- 3.

[Figure]

**Figure S1.** VIC simulations and naturalized streamflow ($10^8$ m$^3$/month). The figure was modified from Yuan et al. (2016).

[Figure]

5  **Figure S2.** Comparison between VIC simulated streamflow from Zhou et al. (2016) without water management model and the naturalized streamflow ($10^8$ m$^3$/month).

**Lijin**

[Figure]

**Figure S3.** The naturalized and observed monthly streamflow ($10^8$ m$^3$/month) at Lijin gauge. The red shaded areas refer to the positive difference between the naturalized and observed monthly streamflow

5   *The problem I have with this paper is the motivation and general approach. Basically, they are analyzing the managed flows as if the management effects were some kind of natural phenomenon. But in fact, the river flows post-management are the result of a set of decisions –either planned, or ad hoc. Those management decisions should be predictable – the irrigated area can be estimated, as can the crop water demands, hence the diversions and return flows. There's a whole branch of water resources*
10   *systems analysis that does just that. So why treat those decisions as a black box, and do a statistical analysis at all? What they have found essentially is that if you look at the season in which the irrigation diversions are made, the river flows go down. Some of that water gets back into the river, later, and perhaps at a different place, and that effect may increase the flows relative to natural (say, somewhere downstream). But all of that should be predictable at some level.*

15   **Response:** We totally agree with the editor for the usefulness of water resources systems analysis. This study presents a very preliminary but quantitative analysis for the human interventions impacts on the relationship between meteorological and hydrological drought, and the hydrological drought characteristics (frequency, duration, severity and drought onset seasonality), where the naturalized streamflow that was basically estimated by using such water resources systems analysis, was used to
20   compare with observed streamflow. As we respond above, the naturalized streamflow has accounted for water consumed by agricultural, industrial and civil sectors, and the water regulated by reservoirs, although all the results are shown as an integral anthropogenic impacts in this study. We are now trying to collaborate with local authority to obtain the water use data in each sectors, and hope to incorporate

those impacts into hydrological model. But we do believe that assessing the integrated human impacts on hydrological droughts as conducted in this study, can also provide useful information for local water resource management. We have added a discussion on the prospect of water resource model in drought analysis as follows:

"Another popular method is to parameterize the human interventions directly in hydrological models (e.g., Wada et al., 2014; Zhou et al., 2016 among many others), where the irrigated area can be estimated for crop water demand, and the diversion and return flows can also be simulated in those models. This modeling framework could be further pushed forward by the availability of Surface Water and Ocean Topography (SWOT) satellite data in the near future, where surface water storage in reservoirs and rivers can be at 250m resolution (Biancamaria et al., 2016)." (P10L10-15)

Another motivation of this study is to explore the seasonal hydrological drought predictability in the anthropocene, which is to our knowledge among the first in the climate-model-based seasonal hydrological forecasting community. Parts of our findings are that "In the anthropocene, the skill for both approaches increases due to dominant influence of human interventions that have been implicitly incorporated by the hydrological post-processing, while the difference between two predictions decreases. This suggests that human interventions can outweigh the climate variability for the hydrological drought forecasting in the anthropocene, and the predictability for human interventions needs more attention." (P1L22-26). So we totally agree with the editor that the management decisions should be predictable to some extent, and considering those predictability would improve the hydrological forecasting in the anthropocene. This study provides a first look on the integrated impacts of human interventions on hydrological drought predictability by using a time series model, and can serve as a benchmark for assessing the physical hydrological model-based approach to fully distinguish the impacts of reservoir regulation, irrigation and groundwater pumping individually in the future. However, the performance of those water management models would also be highly dependent on the data availability, which we believe they are essentially data-driven models in the foreseeable future unless we developed a physical model that fully addresses water and energy balances and the coupling among atmospheric water, surface soil, soil water and groundwater in the anthropocene.

*Also, another concern I have – perhaps not so much with the Yellow River, but with the basins referenced in the Zhang et al, (2014) and Wen et al. (2011) studies that they cite as motivation. The Zhang et al. publication is somewhat obscure, and I could only get the abstract. I did read Wen et al., which is a study of a basin in Australia. In Table 1 of Wen, they give the various water management perturbations to the basin, which include construction of what appears to be a couple of km3 of reservoir storage. There must be an operating policy for those reservoirs, and it must be based on an objective function, presumably having something to do with meeting the irrigation demand. Whether or not the policy deals with instream flows at all isn't clear in the paper. My point is that if you look at the statistics of the reliability of the reservoir system in meeting the irrigation demands, presumably it's*

*higher than without the reservoirs – after all, that's the reason for building reservoirs. But if the operating policy doesn't consider the instream flows, of course eventually enough irrigation will be added to dry up the river. But we don't need a statistical study to tell us that. My concern is that in all of these papers that look at instream drought statistics (including the authors'), that's completely ignored. The situation is slightly different in the Yellow, as I think (I could be wrong) that there isn't currently a lot of storage, so the diversions for irrigation are mostly run of the river. But as I implied above, that could be modeled as well.*

**Response:** We agree with the editor that the statement in last version of the manuscript might cause confusion about the usefulness of building reservoirs. The reservoirs definitely increase the reliability of meeting the irrigation demand, although it may increase the severity of hydrological drought. Moreover, we also agree with the editor the human activities could be modeled. We have revised the statement as follows:

"Similar studies found that the reservoir regulation might reduce the drought severity over upstream areas but increase it over downstream areas over Australian and Chinese catchments (Wen et al., 2011; Zhang et al., 2015), because many reservoirs were built for meeting the irrigation demand more reliably… An alternative approach is to use a land surface hydrological model (Wada et al., 2013; Zhou et al., 2016) or a less complicated water balance model to recover the naturalized streamflow by assimilating the reported water use data." (P2L20-27)

As we respond above, one of the motivations of the study is to quantify the integral impacts of human interventions on hydrological over Yellow River basin, where the naturalized streamflow data generated by a water balance model play an important role. For the situation in the Yellow River, the editor's comment is absolutely correct, the reservoir plays limited role for the mainstream flows at annual scale (please see Figure 4 in the revised manuscript or our response to reviewer #2 below), but their impacts on local small catchments might be significant.

*A final concern I have about the paper is that the ensemble prediction doesn't seem to fit. What was the purpose of including it? Is it to show that if some change in operation was made based on the forecasts, the hydrologic drought statistics would improve? I don't see any argument to that effect. So to me, that part of the paper seems not to fit. I do question the results they show in Figure 7, take for instance for Lijin, which is the sub-basin most affected by diversions for irrigation per their Figure 4. That basin has all kinds of SSI values in the -3 to -9 range, but none of their ensemble members are anywhere close to those values – their smallest forecasts are in the -3 range. The reason of course is that they're using VIC, which (I assume in the version they used) doesn't deal with water management. So they must be forecasting naturalized flows. But who would care about a forecast of naturalized flows? What a management agency needs is a forecast of how much water will be in the river. So we're back to the same thing – to make this paper meaningful, they need a water management model.*

**Response:** I am also a hydro-climate modeler, so I totally agree with the editor that eventually we should have an appropriate physical hydrological model with water management for the hydrological drought forecasting. But to my experiences in modeling and hydrological forecasting, I do not think physical model are ready to resolve everything in the forecasting. The method proposed in this study
5  combines a physical hydrological model (i.e., VIC without human component) with a statistical model (Bayesian model to account for the seasonality of human influences). The disadvantage of this method is that we do not consider the inter-annual variability of human component in the forecasting, which we think it is also very challenging in physical hydrological modeling. Actually we are now collaborating with scientists in USA and China to try to develop a regional climate model that can explicitly account
10  for the surface water storage change and its control on irrigation, in a land-atmosphere coupled mode. The editor's comments and suggestions are very useful for developing such model.

Back to this study and to this comment, we should clarify that human intervention is implicitly considered in Figure 8 (now Figure 11 in the revised manuscript). While for the Figure 7 (now Figure 10 in the revised manuscript), it is to compare with Figure 8 to show the human interventions on
15  hydrological drought forecasting. In addition, Figure 7 itself shows the added value from climate forecasting in a natural world. For the Lijin station, Figures 7 and 8 show a very dry case, where the SSI is extremely low in the observed condition (Fig. 8), but we can see the improvement in the forecasting at its upstream gauges (e.g., Huayuankou, Hekouzhen, etc). The ensemble prediction results show that human intervention can outweigh climate prediction for the hydrological drought forecasting in the
20  anthropocene. Therefore, we would like to keep the ensemble prediction results.

In addition, we have also collaborated with our colleague at PNNL to obtain a set of VIC simulations with and without reservoir regulations and irrigations, and use the difference to represent the human influence, and to correct the hydrological forecasting without considering human interventions. However, it does not perform very well for the hydrological drought forecasting as shown below (Fig.
25  S4). The reasons are multi-folded, but a clear message is that the dynamical-statistical forecasting approach proposed in this study would be a useful intermediate method, before we can collect enough water use data to develop a physically sound model for hydrological drought forecasting.

[Figure]

**Figure S4.** Left: Ensemble SSI prediction correctly by Zhou et al. (2016) VIC simulated human influences on streamflow, as validated by observed SSI. Right: Figure 8 in this manuscript (now Figure 11 in the revised manuscript), where the VIC forecasts are corrected by the Bayesian post-processing method to address the human influence.

*I think the authors need to go back to the drawing board with the entire concept, and take a physically based, rather than statistical approach. As it is currently written, I don't find that the paper provides the reader with many insights into causality, which they could do.*

**Response:** Causality is a very good suggestion, which would be our future target to either collecting
10 more human water use data from different sectors with different purposes, or using a physical hydrological model with water management modules, although large uncertainty might exist without reliable water use data as input. While for this paper, we believe in the naturalized streamflow calculated from a water balance model by the local authorities, which is based on abundant water use datasets that is currently not publically available. So we think the quantification of the human influence
15 on hydrological drought characteristic, and on the drought predictability, could provide some insights for the hydrological drought studies in the anthropocene.

As shown in Figure 5b (Figure 8b in the revised manuscript), the naturalized drought persists a little longer than the observed streamflow at upper gauges, suggesting the positive influence of human intervention for reducing the drought duration. However, the former is basically shorter than the latter at middle and lower gauges, mainly due to intensive human water consumption. We clarified the duration changes in the manuscript as follows:

"For the drought duration in natural conditions, it is generally longer over upper reaches which is again due to a drier climate (Fig. 8b). With human interventions, there is a slight decrease in drought duration for the upper gauges down to Xiaheyan gauge. This suggests that human interventions reduce the persistency of drought over the upper reaches of the Yellow River basin. From Shizuishan gauge (the 6th gauge in Fig. 8b) down to the outlet, the duration of hydrological drought increases by 12%-83% under human interventions." (P7L14-18)

*You used NMME forecasts. Did you perform hydroclimate forecasts using VIC for each model separately and then took the ensemble means? How exactly did you process the NMME data? Readers need more details on that.*

**Response:** Yes, the VIC model was driven by each ensemble members of nine NMME models with a grand ensemble of 99 realizations, and ensemble means were then calculated to evaluate the deterministic forecast skill. In addition, all 99 realizations were also used directly to evaluate the probabilistic forecast skill.

The downscaling process is the same as Yuan (2016). To have this paper self-contained, we have added some descriptions in the revised manuscript as follows:

"Thirdly, a grand ensemble of 99 realizations from eight North American Multimodel Ensemble (NMME; Kirtman et al., 2014) models was used to force hydrological models to generate the

NMME/VIC hindcast dataset (Yuan, 2016). Here, the 1-degree NMME global hindcasts of monthly precipitation and temperature were bilinearly interpolated into 0.25-degree, the interpolated monthly hindcasts for each NMME model were then bias-corrected independently against observations by using the quantile-mapping method (Wood et al., 2002) in a cross-validation mode (i.e., dropping observation
5  and forecast in the target year when generating the climatology), and these bias-corrected monthly hindcasts were finally temporally disaggregated to daily by historical sampling and rescaling (Yuan, 2016)." (P5L21-27)

*You stated seasonal cycle plays a role in drought. (page 5 response time is different for summer and*
10  *winter). How large is the precipitation seasonal cycle?*
**Response:** Thanks for the comment, we have added a figure to show the precipitation seasonal cycle (Figure 2 in the revised manuscript), and have clarified the seasonal cycle as follows:
"Most precipitation over Yellow River occurs in summer season due to the influence of East Asian monsoon, resulting in a strong seasonality of precipitation, with more than 80% of annual precipitation
15  falls within May-September (Fig. 2)." (P3L27-29)

[Figure]

**Figure 2.** Monthly mean rainfall (mm/day) averaged over 1961-2010 for each sub-basin.

naturalized and observed monthly streamflow along the mainstream of Yellow River, although the lag-correlations are again lower for the observed streamflow.

Actually in the last version of the manuscript, we followed the work done by Vicente-Serrano and López-Moreno (2005), and thought the time scale with the maximum correlation is considered as the time scale of SPI that streamflow responds to, i.e., the SPI time scale that has the most similar variations to the SSI. However, we have realized that using "response time" in the manuscript would be very confusing. So, we have removed all "response time" throughout the paper, and have re-written the corresponding text as follows:

**Abstract**—"It is found that human interventions decrease the correlation between hydrological and meteorological droughts, and make the hydrological drought respond to longer time scale of meteorological drought especially during rainy seasons." (P1L14-16 in this response letter)

**Section 3.1**—"Similar to Vicente-Serrano and López-Moreno (2005), the time scale with the maximum correlation is considered as the time scale of SPI that streamflow responds to. For the naturalized streamflow, it responds to 6-12 months SPI over the upper and middle reaches of Yellow River, and to about 4 months SPI over the lower reaches … Except for the Tangnaihai gauge in headwater region, the SPI time scales with the maximum correlation are longer for the observed streamflow than the naturalized streamflow, suggesting that human interventions basically make the hydrological drought respond to longer time scale of meteorological drought … It is found that streamflow responds to shorter time scale of SPI in wet and warm seasons and longer time scale in dry and cold seasons." (P6L4-7, L14-16, L24-26)

**Concluding Remarks**—"Comparison between naturalized and observed SSI at 12 hydrological gauges along the mainstream of the Yellow River basin (the second largest river basin in China with a drainage area of $7.52 \times 10^5$ km$^2$) shows that human interventions decrease the correlation between hydrological and meteorological droughts, and make the hydrological drought respond to longer time scale of meteorological drought especially during rainy seasons." (P9L14-17)

*Further, the notion of "nonlinear response" of hydrological drought to meteorological drought is a bit vague in the discussions. The rainfall-runoff process is by itself "nonlinear" and lagged in time, at least at short time scales. If the word "response" refers to the rainfall-runoff process (at any time scale), the research here should try to find out what exactly human interventions did to that process. Reduction in streamflow volume (i.e. significant amount of consumptive use)? Longer lag times (delayed release for flood control)? If the lag times become longer, should this be considered in the forecast post-processing procedure? (For example, a time series based procedure that looks at a prior history instead of just the current month.)*

**Response:** We have also removed "nonlinear response" in the revised manuscript given that the focus of this work is not to investigate the lag-correlation for forecasting. We have plotted the annual cycle of

naturalized and observed streamflow in Figure 3 to show the effect of human interventions, and revised the section "2.1 study domain and hydroclimate observation data" as follows:

"In general, human interventions decrease streamflow over upper and middle reaches of Yellow River during rainy season while increase it during dry season (Figs. 3b-3h). This suggests that reservoirs in the upper and middle reaches of Yellow River store rain water in wet season and distribute it in the remaining time of the year according to the need, which is similar to regulations in other parts of the world (Wada et al., 2014). Actually, Figure 4a shows that the annual mean observed streamflow at upper reaches can be higher than the naturalized streamflow during dry years due to the reservoir water release (e.g., years 2000, 2002, 2006 and 2010 for Lanzhou gauge). Over the lower reaches, the observed streamflow is significantly lower than the naturalized streamflow during wet season due to heavy water consumption (Figs. 3i-3l), but the former is close to the latter during dry season because of no significant water consumption or reservoir management. (Figs. 4c-4d)" (P4L6-13)

[Figure]

**Figure 3.** Monthly mean naturalized (blue) and observed (red) streamflow ($10^8$ m$^3$) averaged over 1961-2010 for 12 hydrological gauges located from upper to lower mainstream of the Yellow River.

*For the same reasons, more information on the water regulation practices in the study area is needed for a (much) better understanding of the impacts and differences found in the results. For example, reservoirs may store rain water from wet season and distribute it in the remaining time of the year according to the need. How much of the streamflow water is being regulated in the Yellow River basin*

*(e.g. reservoir capacity relative to the annual total inflow) and for what purposes? How much of the streamflow is being modified (in both absolute and relative senses)?*

**Response:** We thank for the comment. We have now collected annual statistics for the reservoir storage change during 1998-2010, but failed to obtain the monthly data. Based on the data available, we have added Figure 4 to show the interannual variations of naturalized and observed streamflow and the reservoir storage change, and we have revised the manuscript as follows:

"Figure 4 also shows that the magnitudes of reservoir storage changes are quite small as compared with streamflow. In fact, the mean absolute changes of reservoir storage during 1998-2010 are about 14%-38% and 12%-14% of observed and naturalized streamflow, respectively. This suggests that other human interventions, such as direct withdrawal of surface water for agricultural, industrial and civil consumptions, account for a large part of streamflow variations over Yellow River." (P4L14-18)

[Figure]

**Figure 4.** Annual mean naturalized (blue) and observed (red) streamflow ($10^8$ m$^3$), and reservoir storage change ($10^8$ m$^3$, negative green values represent reservoir water distribution) accumulated within four selected sub-basins (from headwater down to the gauge) during 1998-2010.

*Fig. 4 helps to understand the scenario but direct comparisons between observations and naturalized values (in seasonal cycles and annual totals) can help explain what happened in Fig. 4 in a much better way. I guess the observed SSI in Fig. 4 is calculated against observed flow climatology and naturalized SSI against naturalized flow, right? (Please clarify.) If so, the comparisons between the two do not*

*reveal the difference between the observed and naturalized climatologies, e.g. reduced total flow volumes or lagged peak times.*

**Response:** Observed SSI in Fig.4 is not calculated against observed flow climatology. Actually both naturalized and observed SSI are calculated against the naturalized flow climatology, so they can be compared to detect the effect of human interventions on hydrological drought. Seasonal cycle of original values are now shown in Figure 3 (please see our response above) to support the SSI analysis.

*Specific information on the local water management and water use practices is always helpful in understanding the findings and their implications across similar areas in other parts of the world (Wada et al., 2014). The study could be significantly stronger if more specific water management information is provided and related to the research findings.*

**Response:** Thanks for the comment. Two figures regarding the seasonal cycle of monthly naturalized and observed streamflow, and the annual mean streamflow and reservoir storage change have been added into the revised manuscript. Please see our responses above.

*P. 5, L. 13: nonlinearly -> nonlinear*

**Response:** Revised as suggested.

*Fig. 1: The map needs to show at least the Yellow River and its main tributaries (thicker lines for the main stream) under this study. Replace the political boundaries with sub-basin boundaries (keep the coast lines).*

**Response:** We have revised Figure 1 as suggested.

[Figure]

**Figure 1.** Locations of hydrological stations over the Yellow River basin. Shaded areas are regional mean annual rainfall (mm/day) averaged during 1961-2010.

*Fig. 4: SSI at what time scale? 1-month? Subplots are too small and better if they are rearranged into multiple columns.*

**Response:** Fig. 4 (Fig. 7 in the revised manuscript) has been replotted to show the panels in two columns. The SSI is at 1-month time scale, and it has been clarified in the revised figure caption:

[Figure]

**Figure 7.** Time series of naturalized (blue) and observed (red) 1-month Standardized Streamflow Index (SSI) for five selected hydrological gauges. The horizontal black lines represent the threshold of -0.8 for drought conditions.

---

## Author Response (AR2)

[revised manuscript text omitted]

Email: yuanxing@tea.ac.cn
Tel: +86-10-82995385
http://www.escience.cn/people/yuanxing

May 7, 2017

Prof. Dennis Lettenmaier
Special Issue Editor
Hydrology and Earth System Sciences
RE: manuscript #hess-2016-592

Dear Prof. Lettenmaier,

Thank you for your kind decision letter on our manuscript entitled "Understanding and seasonal forecasting of hydrological drought in the anthropocene" (hess-2016-592). We have carefully considered the third reviewer's comments and incorporated them into the revised manuscript to the extent possible. Major changes include clarifying the naturalized streamflow data and the groundwater overdraft, considering the non-stationarity by using transient 30-year climatology for the calculation of SPI/SSI and the analysis, and updating Figs. 5-9 and corresponding results. The non-stationarity generally amplifies the human influence on hydrological drought duration and severity over the middle and lower reaches of Yellow River, while reduces the influence on hydrological drought frequency. We hope that you find the revised manuscript and the response to the reviews acceptable to *HESS*.
The detailed responses to the comments are attached.

We appreciate the effort you spent to process the manuscript and look forward to hearing from you soon.

Sincerely yours,

Xing Yuan

**Responses to the comments from Reviewer #3**

We are very grateful to the reviewer for the positive and careful review. The thoughtful comments have helped improve the manuscript. The reviewer's comments are italicized and our responses immediately follow.

*This paper aims to understand the effect of human activities on the occurrence of hydrological droughts. By making use of this understanding and the seasonal climate forecasts from multiple GCMs, the paper also aim to demonstrated potential of performing seasonal forecasting of hydrological droughts. I think the authors did a reasonable job on highlighting the importance of accounting for human activities in*

10 *understanding and forecasting hydrological droughts, especially in Yellow River basin in North China, where human activities have significant effects on hydrological processes.*

**Response:** Thanks for the positive comments.

*After reading through the paper, the authors seem to have ignored a couple of huge factors, that is, the*

15 *groundwater overdraft can cause significant leakage of surface water to groundwater, and Yellow river is famous for its changing geomorphology due to heavy sediments from the Loess Plateau carried downstream, which would elevate the riverbed over the years, making streamflow time series non-stationary, i.e., even without human activities, hydrological response to the same meteorological forcing could change over time. I don't think the authors have considered those factors adequately.*

20 **Response:** We greatly thank the reviewer for the positive and helpful comments. After consulting the local authorities of Yellow River, we have also improved our understanding of the naturalized streamflow data and the hydrological condition over Yellow River. Yes, the riverbed leakage due to groundwater overdraft is significant over the middle and lower reaches of Yellow River. However, the calculation of naturalized streamflow has already considers such surface-subsurface water exchanges

25 (please see our response to the comment on naturalized streamflow data below). Therefore, we believe our analysis based on the naturalized and observed streamflow data has considered the groundwater overdraft effect, and we have clarified it in the revised manuscript.

For the seasonal hydrological forecasting part, we acknowledge that we use a VIC model that does not consider the groundwater overdraft effect. Actually the VIC model used here does not have a

30 groundwater component, but it does calculate the subsurface runoff (baseflow) by using a conceptual ARNO formulation (Todini, 1996) which is based on a three-parameter nonlinear recession curve. In the natural condition, the VIC model and the corresponding routing model (which routes both surface runoff and baseflow) are calibrated to reproduce the naturalized streamflow, where the effect of

riverbed leakage processes is implicitly considered in order to capture the dynamics of naturalized streamflow. For the observed conditions with human interventions, the effect of groundwater overdraft is also implicitly considered in the hydrological post-processing procedure that transfer the VIC simulated streamflow to those comparable to the observed streamflow (which includes the groundwater overdraft effect) based on the Bayesian merging technique. Therefore, we believe the seasonal hydrological forecasting has also considered the groundwater overdraft effect to some extent, although we totally agree a more physically-based hydrological model is needed for future study (please see discussion at the end of the manuscript).

The effect of sediment accumulation on streamflow is limited as compared with other factors such as irrigations, industrial and civil water uses. In fact, the sediment accumulation is decreasing in recent years due to the Grain for Green ecological conservation project.

However, there are many other reasons that may lead to the non-stationarity of streamflow, e.g., climate change, land use/land cover changes, and human water consumption. To address this issue, we have augmented the calculation of SSI that based on the 50-year (1961-2010) climatology with that based on a transient climatology (i.e., 30-year moving window), re-calculated the results from Figure 5 to Figure 9, and revised the manuscript as follows:

"Due to large water consumptions over the middle and lower reaches, there are 118%-262% increases in the hydrological drought frequency and up to eight-fold increases in the drought severity, the drought duration increases by 21-99%, and the drought onset becomes earlier. The non-stationarity due to anthropogenic climate change and human water use basically decreases the correlation between meteorological and hydrological droughts, reduces the effect of human interventions on hydrological drought frequency while increase the effect on drought duration and severity." (P1, L15-20)

"Both the 50-year (1961-2010) data and a series of 30-year transient climatology (e.g., 1961-1990 climatology was used for year 1975, 1962-1991 climatology was used for 1976, and so on) were used to calculate the stationary and non-stationary SPI/SSI values respectively." (P5, L25-27)

"By considering the non-stationarity, the correlations decrease generally from Longmen gauge (Fig. 5h) to the downstream areas, and the decrease are more obvious for the observed streamflow and for the longer timescale of SPI. This suggests that the relation between meteorological drought and hydrological drought over lower reach of Yellow River is not stationary, anthropogenic climate change and human interventions add more nonlinearity to the propagation from meteorological to hydrological droughts." (P7, L8-12)

"Again, non-stationarity basically weakens the relation between meteorological and hydrological drought over the lower reaches, regardless of seasons." (Figure 6; P7, L19-20)

"The results based on the 50-year climatology generally have lower SSI in 1990s-2000s and higher SSI in 1960s-1970s (now shown), suggesting there are decreasing trends in streamflow over Yellow River basin that is consistent with previous studies (e.g., Piao et al., 2010)." (Figure 7; P7, L28-31)

"As compared with the results based on a constant climatology (cyan and pink bars in Fig. 8), the non-stationarity generally reduces the human influence on drought frequency, but increases the human impact on drought duration and severity." (P8, L14-16)

"With human interventions, the hydrological drought onset becomes earlier, no matter using transient or constant climatology (Fig. 9c-d)." (P8, L23-24)

*Also, I echo with editor's concern on how naturalized streamflow is computed. Based on the paper, it is hard to figure how naturalized streamflow is truly computed. I wonder if the authors have access to the paper version of Water Resources Bulletins for all years between 1961-2010, because the website http://www.yellowriver.gov.cn/zwzc/gzgb/ has only annual observed streamflow, annual water withdrawals and consumptions for 1998 to 2015. Further the bulletins have only annual numbers, not the monthly numbers used in the study, so I wonder how they are obtained. Besides, I don't see direct naturalized data from the website. So the authors basically estimated the natural streamflow themselves by adding water consumptions by different sectors. I don't know if such a simple addition/substraction is proper because of the complex interactions between surface water and subsurface water and changing geomorphology, as mentioned above.*

**Response:** We thank the author for the scrutinization! We actually obtained the monthly data from local authority of Yellow River, and was told to use the "Water Resources Bulletins" as an official reference to the data. But we have now realized that the publically available data in the bulletin are annual mean data, although they do have monthly data in the internal bulletin. After consulting the local authority about the procedure of calculating naturalized streamflow in details, we have now clarified the data as follows:

"Both the natural and observed streamflow datasets at monthly time scale during 1961-2010 were provided by local authority of the Yellow River. The naturalized streamflow ($W_{nat}$) was calculated as follows:

$$W_{nat}=W_{obs}+W_{irr}+W_{idu}+W_{civ}+W_{div}+W_{res},$$

where $W_{obs}$ is the observed streamflow; $W_{irr}$ is surface water consumed in irrigation, i.e., irrigated surface water that is transpired from crop, evaporated from bare ground and river/channel water surface, absorbed into soil (through infiltration, percolation and recharge to shallow groundwater), leaked from river/channel bed to groundwater during transportation, so the $W_{irr}$ is not simply those withdrew from

river, while it is actually the surface water consumption in the processes of irrigation that already considers the flow return to the river; similarly, $W_{idu}$ and $W_{civ}$ are surface water consumed by industrial and civil sectors, by considering the waste water returned to the river; $W_{div}$ is the interbasin water diversion, and $W_{res}$ is the water regulated by reservoirs. To account for the non-stationarity both from anthropogenic climate change (Milly et al., 2008, 2015) and land use/land cover changes (Villarini et al., 2009; Zhang et al., 2011), the variable runoff coefficient (annual runoff divided by annual precipitation) and more physically based methods that consider the soil conservation (e.g., Grain for Green project over Loess Plateau), effect of groundwater overdraft on surface water, and evaporation loss from reservoirs or channels, are used to correct the naturalized streamflow. Such correction is critical over Yellow River due to significant climate change and human interventions (Zhang et al., 2011). For example, small, medium and heavy rainy days decreased by 20-26% over the areas between Hekouzhen and Longmen gauges in 1980s as compared with those in 1950s and 1960s, with duration of extremely heavy rainfall decreased from 0.82 day to 0.38 day. The annual streamflow over the middle reaches of Yellow River could decrease by 19% with the same annual precipitation due to ecological conservation, and could decrease by 47% due to the riverbed leakage induced by groundwater overdraft. The corrections for the annual mean streamflow over the areas from Tangnaihai to Lanzhou, and down to Hekouzhen, Longmen, Sanmenxia, Huayuankou and Lijin are $3.16\times10^8$, $0.80\times10^8$, $6.17\times10^8$, $10.99\times10^8$, $9.23\times10^8$ and $2.61\times10^8$ m$^3$, respectively. To our knowledge, this is the most comprehensive estimation of naturalized streamflow over Yellow River basin so far, based on abundant data from different sectors. Yuan et al. (2016) used the naturalized streamflow to calibrate the Variable Infiltration Capacity (VIC) land surface hydrological model (Liang et al. 1996) during 1961-1981, when the human interventions were supposed to be limited. The naturalized streamflow was then compared with VIC simulations during the validation period 1982-2010, and it is found that the naturalized streamflow agreed with VIC simulations quite well with Nash–Sutcliffe efficiency (NSE) varied between 0.71-0.91 (Figure 4 in Yuan et al., 2016)." (P4, L1-30)

*Except for my concerns on how to account for full effect of human activities and non-stationarity, I have no big problem on the design on figuring out the relational between meteorological drought and hydrological drought, nor do I question hydrological forecasting experiments. I tend to agree that human activities worsen the hydrological droughts. The questions the authors need to address if those quantitative numbers measuring the effects of human activities are truly reliable. In summary, I think that the authors still to clarify those points to make sure all effects of human activities are*

*comprehensively considered and naturalized streamflow are truly reflective of natural conditions. My recommendation is to return for further revision.*

**Response:** Thanks for the comments. For the groundwater and non-stationary issues, please see our responses above. The clarification of natural conditions is also presented above. By adding analysis of non-stationarity in the first part of the paper according to the reviewer's helpful suggestion, we have now obtained more objective numbers with a certain range.

*Some specific comments:*

*P1, Abstract, line 14-15: Something is missing in this phrase: "... make the hydrological drought respond to longer time scale of meteorological drought especially during rainy seasons". Especially what?*

**Response:** Human interventions increase the response time of hydrological drought to meteorological drought, especially during rainy seasons. We have removed "especially during rainy seasons" (P1, L15)

*P1, line 27: I am not sure if "recursive" is the right word for describing an event occurred repeatedly. According to www.thefreedictionary.com, "Recursive" is term "relating to a repeating process whose output at each stage is applied as input in the succeeding stage." Estimation of future soil moisture can be recursive as future soil moisture can use current soil moisture as input. This doesn't apply to droughts as future drought cannot use current drought condition as input.*

**Response:** We have removed the word "recursively". (P1, L37)

*P3, line 34-35: see my comments above. I feel how naturalized streamflow is obtained needs further clarification.*

**Response:** Thanks for the suggestion. We have augmented the clarification of the naturalized streamflow. Please see our response above.

*P4, line 10-15: Again, I would like to point out that the lower natural streamflow can also be due to riverbed leakages because of the overdraft of groundwater and of changing geomorphology due to sediment accumulation downstream.*

**Response:** Thanks for the comment. We have revised it as follows:

"Over the lower reaches, the observed streamflow is significantly lower than the naturalized streamflow during wet season due to heavy water consumption, and riverbed leakages because of groundwater overdraft and possible geomorphology change caused by sediment accumulation" (P5, L5-7)

*P5, line1-24: here, the streamflow non-stationarity question needs to be addressed.*

**Response:** The reviewer raised an interesting yet very challenging issue: how to address non-stationarity in the seasonal hydrological forecasting? However, to my knowledge, there is currently

almost NO publication that addresses this issue. The reasons are multifold. First, seasonal hydrological forecasting is to try to forecast hydrological variables (e.g., streamflow, soil moisture) in the next few months (e.g., 3-6 months), where the hydrological model is initiated with current hydrological conditions, and is forced by climate model predictions that are always assumed that the climate within the next 3-6 months is stationary. Second, although there might be unexpected non-stationarity such as UNEXPECTED land cover change in the next 3-6 months, researchers tend to focus more on how to improve the climate and hydrological predictive skill by investigating the sources of hydro-climate predictability, which have been shown a steady improvement during the past 30 years (e.g., ENSO teleconnection, climate-model-driven hydrological forecasting, high-resolution forecasting). In my opinion, the non-stationarity in the seasonal hydrological forecasting could be addressed in the adaptation stage, by focusing on the science questions such as how does human respond to a changing environment (both climate and hydrology, and etc.), how to add human dimension in the seasonal hydrological forecasting scenarios.

Back to out our manuscript, the non-stationarity was actually already considered to some extent. We have now clarified it as follows:

"To account for the non-stationarity, the hydrological post-processing was carried out by using observed streamflow during 1982-2010 in a cross-validation mode (Yuan, 2016), and the corresponding SSI was also calculated by using the concurrent hindcast period (i.e., 1982-2010). It should be pointed out that the non-stationarity could be reduced further in a real-time forecasting mode because of the gradual use of concurrent climate and hydrology information for the calibration and initialization of hydrological model for the (seasonal) hydrological forecasting in the next 3-6 months." (P6, L21-26)

---

## Author Response (AR3)

Xing Yuan
Professor/Dr
Institute of Atmospheric Physics
Chinese Academy of Sciences
Beijing 100029, China
Email: yuanxing@tea.ac.cn
Tel: +86-10-82995385
http://www.escience.cn/people/yuanxing

September 29, 2017

Prof. Dennis Lettenmaier
Special Issue Editor
Hydrology and Earth System Sciences

RE: manuscript #hess-2016-592

Dear Prof. Lettenmaier,

Thank you for your kind decision letter on our manuscript entitled "Understanding and seasonal forecasting of hydrological drought in the anthropocene" (hess-2016-592). We have carefully considered the third reviewer's comments and incorporated them into the revised manuscript to the extent possible. We hope that you find the revised manuscript and the response to the reviews acceptable to *HESS*.

The detailed responses to the comments are attached.

We appreciate the effort you spent to process the manuscript and look forward to hearing from you soon.

Sincerely yours,

Xing Yuan

**Responses to the comments from Editor**

*Please respond to the reviewer's comment. I think the reviewer's main complaint is that you've basically dismissed issues with the naturalized flows, which underlie your analysis. As the reviewer points out, it's pretty much impossible to know how well or poorly the estimates reproduce actual natural conditions. The reviewer points out the issue of interactions among the management effects for instance.*

*I suggest what you do is write a couple of sentences acknowledging that the analysis is dependent on the naturalized flows, and state some of the limitations (don't appear to dismiss them). But i don't see that there is much more you can do. Once you've made that change, I'm prepared to accept the paper.*

**Response:** Thanks for the positive comment and helpful suggestion. We have acknowledged the uncertainties by adding a few comments in the revised manuscript. Please see our response below.

**Responses to the comments from Reviewer #3**

*This latest revised manuscript has responded to all the comments of the last version. Particularly, the authors answered the questions related to my concerns about non-stationarity caused by natural elements such as sedimentations of the Loess Plateau on downstream channels and human elements such as over-pumping of groundwater on surface-subsurface interactions. They also provided a reply to the concern about how naturalized streamflow was created. I have no problem with the writing of the paper. It is mostly well presented.*

**Response:** Thanks for the positive comments.

*Reading those replies, however, I found that the authors are too eager to dismiss the issues raised in previous comments, instead of acknowledging and opening up to the limitations of the study in the revision. For example, naturalized streamflow has obvious limitations as they are derived under very strong assumptions. Based on what I read from the manuscript, the way the naturalized streamflow is derived in this study is simple summations and deductions and it did not consider the interactions different elements. Nobody knows for sure how naturalized streamflow compared to streamflow under the natural conditions. This is a given fact no matter how the creator of the naturalized streamflow data defends it. Granted, naturalized streamflow is probably the best information available to us, but there is considerable uncertainty associated with it, and the uncertainty is probably not easy to quantify. I think the authors should just acknowledge this limitation explicitly in the text where naturalized streamflow first appeared and in the conclusion sections.*

*Overall, I recommend further revision.*

**Response:** Thanks for the positive comments. We have acknowledged uncertainty in the revised manuscript as follows:

**Section 2.1**

"Definitely, there are uncertainties in the naturalized streamflow which is difficult to quantify because of absent "real" streamflow under natural conditions over Yellow River." (P4, L21-23)

**Concluding Remarks**

"However, these estimations are heavily based on the quality of naturalized streamflow data. Current procedure of generating naturalized streamflow is basically data driven, where the interaction among different elements is not explicitly considered. In the future, more sophisticated method such as assimilating those precious data into a physical hydrology model that explicitly considers surface water-groundwater interactions and human influences, is necessary for a more robust estimation of naturalized streamflow. Multisource satellite retrieval data (e.g., GRACE terrestrial water storage change, SMAP soil moisture, and MODIS

evapotranspiration) could also be a useful complement to in-situ data and hydrological modelling for the estimation." (P10, L15-21)